Resource

# EmbryoNet: using deep learning to link embryonic phenotypes to signaling pathways

Daniel Čapek [1,2,5], Matvey Safroshkin[3,5], Hernán Morales-Navarrete [1,2,4,5], Nikan Toulany [1,2], Grigory Arutyunov[3], Anica Kurzbach[1], Johanna Bihler[2], Julia Hagauer[2], Sebastian Kick[2], Felicity Jones[2], Ben Jordan[1] & Patrick Müller [1,2,4] ✉

Evolutionarily conserved signaling pathways are essential for early embryogenesis, and reducing or abolishing their activity leads to characteristic developmental defects. Classification of phenotypic defects can identify the underlying signaling mechanisms, but this requires expert knowledge and the classification schemes have not been standardized. Here we use a machine learning approach for automated phenotyping to train a deep convolutional neural network, EmbryoNet, to accurately identify zebrafish signaling mutants in an unbiased manner. Combined with a model of time-dependent developmental trajectories, this approach identifies and classifies with high precision phenotypic defects caused by loss of function of the seven major signaling pathways relevant for vertebrate development. Our classification algorithms have wide applications in developmental biology and robustly identify signaling defects in evolutionarily distant species. Furthermore, using automated phenotyping in high-throughput drug screens, we show that EmbryoNet can resolve the mechanism of action of pharmaceutical substances. As part of this work, we freely provide more than 2 million images that were used to train and test EmbryoNet.

Early development is governed by a handful of signaling pathways that balance tissue growth, differentiation and morphogenesis[1–3]. Given their important roles in controlling cell identity and behavior, misregulation of signaling pathways in adult tissues can induce the formation of tumors with embryo-like properties, defective cell proliferation and migration[4,5].

During zebrafish development, seven major signaling pathways orchestrate the formation of the body plan. Bone morphogenetic protein (BMP), retinoic acid (RA), Wnt, fibroblast growth factor (FGF) and Nodal pattern the germ layers and regulate the formation of the orthogonal anterior–posterior and dorsal–ventral axes; Sonic hedgehog (Shh)

and planar cell polarity (PCP) signaling, in turn, control the elongation and morphogenesis of the body axis and later shape the formation of tissues[2,3,6–8]. The ligands activating these signaling pathways are dynamically expressed from specific source tissues in the embryo (Fig. 1a and Supplementary Note 1). Loss of activity in any of these pathways causes characteristic patterning defects, which, however, can be difficult to distinguish (Fig. 1b and Supplementary Videos 1–8). For example, both Nodal and Shh mutants have cyclopic eyes (Fig. 1b and Supplementary Videos 6 and 7), but the defect in Nodal mutants is caused by an early lack of mesoderm[9], whereas cyclopia in Shh mutants is caused by a late defect in midline patterning[10]. Furthermore, while

[1]Systems Biology of Development, University of Konstanz, Konstanz, Germany. [2]Friedrich Miescher Laboratory of the Max Planck Society, Tübingen, Germany. [3]Computer Vision Studio, Tübingen, Germany. [4]Centre for the Advanced Study of Collective Behaviour, Konstanz, Germany. [5]These authors contributed equally: Daniel Čapek, Matvey Safroshkin, Hernán Morales-Navarrete. ✉e-mail: patrick.mueller@uni-konstanz.de

**Fig. 1 | The CNN EmbryoNet robustly identifies molecular defects based on phenotype data. a**, Simplified schematic of signaling domains during zebrafish development projected onto an early embryo. **b**, Schematic drawings of zebrafish embryo phenotypes. −BMP loss of function causes reduced and often curled tails, +RA gain-of-function embryos lack head structures and have shortened tails, −Wnt leads to enlarged heads and shortened tails, −FGF causes loss of mesoderm and tail tissue, −Nodal embryos lack mesoderm and have cyclopia, −Shh embryos frequently have mispatterned somites and cyclopia, and −PCP leads to a shortened and widened body axis, manifested, for example, by shorter somites. **c**,**d**, Treatment with the chemical Nodal inhibitor SB-505124 caused specific phenotypes that were not yet apparent at sphere stage (**c (i)**), but which were clearly visible at segmentation stages (**c (ii)**); $n = 36$. The inhibitor treatment (**c (iii)**) phenocopied the MZ*oep* (**d**) mutant, and both phenotypes were robustly identified by EmbryoNet; $n = 58$. **e**,**f**, Schematic overview of the

neural network architecture with convolutional (Conv) layers shown in blue. Stack sizes after each image filter are illustrated in **e**, whereas **f** details the filters of the network. Relu, rectified linear unit. **g**–**i**, EmbryoNet correctly classified embryos in a mixed population. **g**, Experimental set-up. Embryos at the one-cell stage were injected with mRNA encoding the Nodal inhibitor Lefty1 and Alexa647-labeled dextran (magenta), mRNA encoding the BMP inhibitor Chordin and Alexa488-labeled dextran (green), or were left uninjected (wild type) and then imaged. Black bounding boxes indicate the class Unknown; green indicates −Nodal; red indicates −BMP; white indicates Normal and magenta indicates Dead. **h**, At the sphere stage, EmbryoNet labeled the phenotypes as Unknown. Dextran-labeling shows the applied treatment. **i**, During segmentation stages the Normal, −BMP and −Nodal samples were correctly identified by EmbryoNet. The classification is in accordance with the dextran colors; $n = 85$. Scale bar, 500 μm.

misregulation of the BMP, Wnt, RA, FGF and PCP signaling pathways leads to specific defects, for example, an enlarged head in the case of Wnt mutants[11,12], all of these mutants also have malformed shortened tails[13–17] (Fig. 1b and Supplementary Videos 2–5, 8). Thus, the phenotypes caused by changes in the activity of different signaling pathways can be easily confused by even the most experienced developmental biologists. Automated and unbiased phenotyping based on a multitude of morphological features would overcome this challenge. Such an approach would rapidly link phenotypes arising from genetic defects, mutants identified in forward and reverse genetic screens, or treatment with small-molecule inhibitors to the relevant signaling pathway. Automated phenotyping of morphological defects would thus enhance both the speed and accuracy of biological and pharmaceutical discovery.

Advances in deep learning approaches[18] have brought unprecedented breakthroughs in numerous fields ranging from bioimage analysis and visual object recognition[19] to protein structure prediction[20–23] and earth system science[24]. Deep learning approaches perform exceptionally well in decoding the content of images[25,26], and convolutional neural networks (CNNs) in particular have been extensively used for bioimage restoration[27], cell detection and classification[28] and bioimage data segmentation[29]. Recent studies have also used machine learning approaches to examine embryonic phenotypes[30–36], but these approaches were limited to staging, segmentation and classification of specific embryos and organs without being able to uncover the molecular basis of morphological alterations.

Here, we introduce a deep learning approach, EmbryoNet, that can detect specific defects linked to the seven major vertebrate signaling pathways by automated phenotyping. EmbryoNet was trained on more than 2 million images, comprising thousands of trajectories of normally developing and signaling-defective zebrafish embryos. We found that EmbryoNet identified phenotypes more precisely and often long before human evaluators could detect them. By using the accelerated phenotype classification of EmbryoNet in an automated drug screen, we identified novel teratogenic effects caused by Food and Drug Administration (FDA)-approved substances not previously associated with the regulation of developmental signaling pathways. Finally, we show that EmbryoNet can identify signaling defects in evolutionarily distant species, demonstrating the generalizability of our approach.

## Results

### Identification of signaling defects in zebrafish embryos

To test whether deep learning approaches can be used for the automatic classification of complex phenotypes caused by the loss of signaling pathways in zebrafish, we combined high-throughput imaging with specific drug-mediated loss-of-function approaches. We started with proof-of-concept experiments focused on Nodal signaling because both the signaling pathway constituents and their functions during the first day of zebrafish embryogenesis are well described[37] (Supplementary Note 1 and Fig. 1a). In addition, a specific small molecule targeting the ATP-binding pocket of the receptor kinase is available[38] (Supplementary Note 2), facilitating the rapid acquisition of defined developmental phenotypes from a large number of embryos. Indeed, the inhibitor SB-505124 clearly recapitulated the known loss-of-function phenotypes of Nodal signaling pathway components[39], that is, cyclopia and loss of all endoderm and head–trunk mesoderm (Fig. 1a–c and Supplementary Video 6). We then acquired bright-field movies of SB-505124-treated and untreated embryos in random orientations, comprising a total of 342,559 images between 2 and 26 hours post-fertilization (h.p.f.). A modified version of the ResNet18 CNN that includes a timestamp of the images (Fig. 1e,f and Methods)[40] trained with these datasets robustly and correctly identified normal and Nodal-defective embryos, independent of their orientation and whether small molecules (SB-505124) or mutants (maternal zygotic *oep* mutants, MZ*oep*[41]) were used to create Nodal loss-of-function phenotypes (Fig. 1c,d and Extended Data Fig. 1).

We next extended this approach to the seven major signaling pathways that control early development: BMP, RA, Wnt, FGF, Nodal, Shh and PCP (Fig. 1a,b). Using a chemical genetics approach with specific signaling pathway modulators[38,42–45] (Supplementary Table 1 and Supplementary Note 2), we created a dataset of more than 2 million images with loss-of-function (or gain-of-function in the case of RA) phenotypes (Supplementary Videos 1–8). The dataset was manually annotated by curators who were informed about the treatment of each embryo. The curators assigned classes appropriate for each treatment (that is, −BMP, +RA, −Wnt, −FGF, −Nodal, −Shh and −PCP) at the developmental timepoint when the phenotype first became apparent for a given embryo. The class Unknown was assigned when an image did not contain sufficient information for classification, and the class Dead was assigned if an embryo disintegrated over the course of development. In addition, each image was assigned a timestamp for classification (Fig. 1f). This high-confidence dataset was then used to train a large-scale CNN with accelerated graphics processing unit computing (Methods and Supplementary Tables 2 and 3).

To correct for potential classification errors, we introduced a model transition logic based on our knowledge of developmental changes: in very early embryos, phenotypic differences are not yet apparent because signaling changes result in morphological changes only at later stages[1–3]. These early embryos, characterized by the phenotype class Unknown, can then transition to another phenotype class (−BMP, +RA, −Wnt, −FGF, −Nodal, −Shh and −PCP) and can also change to Dead at any point in time (Extended Data Fig. 1c). However, certain transitions, for example, from Dead to Normal, are not possible. We therefore assigned a cost to every state transition in an individual embryo track and scored the cost for different models. The transition sequence that achieved the lowest cost was selected for classification. This approach yielded a classification performance of 89%. The deep learning-based classification network, termed EmbryoNet, was able to robustly identify the loss-of-function phenotypes caused by orthogonal approaches such as the injection of messenger RNAs (mRNAs) encoding the Nodal and BMP pathway inhibitors Lefty1 and Chordin, respectively (Fig. 1g–i, Supplementary Note 2 and Extended Data Fig. 1f,g). EmbryoNet's algorithms for the detection, tracking, manual and automatic classification of embryos are available as easy-to-use, modular and open-source graphical user interface (GUI) software (Extended Data Fig. 1e; http://github.com/mueller-lab/EmbryoNet and http://embryonet.uni-konstanz.de).

### EmbryoNet is proficient, fast and accurate

To evaluate EmbryoNet's performance, we tested its classification speed and accuracy in competition with human assessors. We generated stacks of 98 embryo images, representing the full spectrum of our phenotype classes. These images had not been used previously for the training of EmbryoNet, and information about the specific treatment of each embryo was not disclosed to the assessors.

Random guessing resulted in an accuracy of 9% (F-score = 0.09; Fig. 2a and Supplementary Table 4). The images were then classified by non-experts. These 55 teams, each consisting of two assessors with a biology background, received 1 day of developmental biology training with a focus on developmental defects caused by modulated signaling in zebrafish (Supplementary Videos 1–8, Fig. 2b and Extended Data Figs. 2 and 3). We encouraged the assessors to discuss the phenotypes to make the best classification choice. On average, non-experts confidently identified the class Dead but identified signaling defects with an overall accuracy of only 53% (F-score = 0.52; Fig. 2b, Extended Data Fig. 2 and Supplementary Tables 5 and 6), even when temporal information about the developmental stage was provided (accuracy of 54%, F-score = 0.52; Fig. 2c, Extended Data Fig. 3 and Supplementary Tables 5 and 7). The images were next classified by an expert assessor, an experienced developmental biologist with several years of relevant background in early zebrafish embryogenesis. The expert confidently

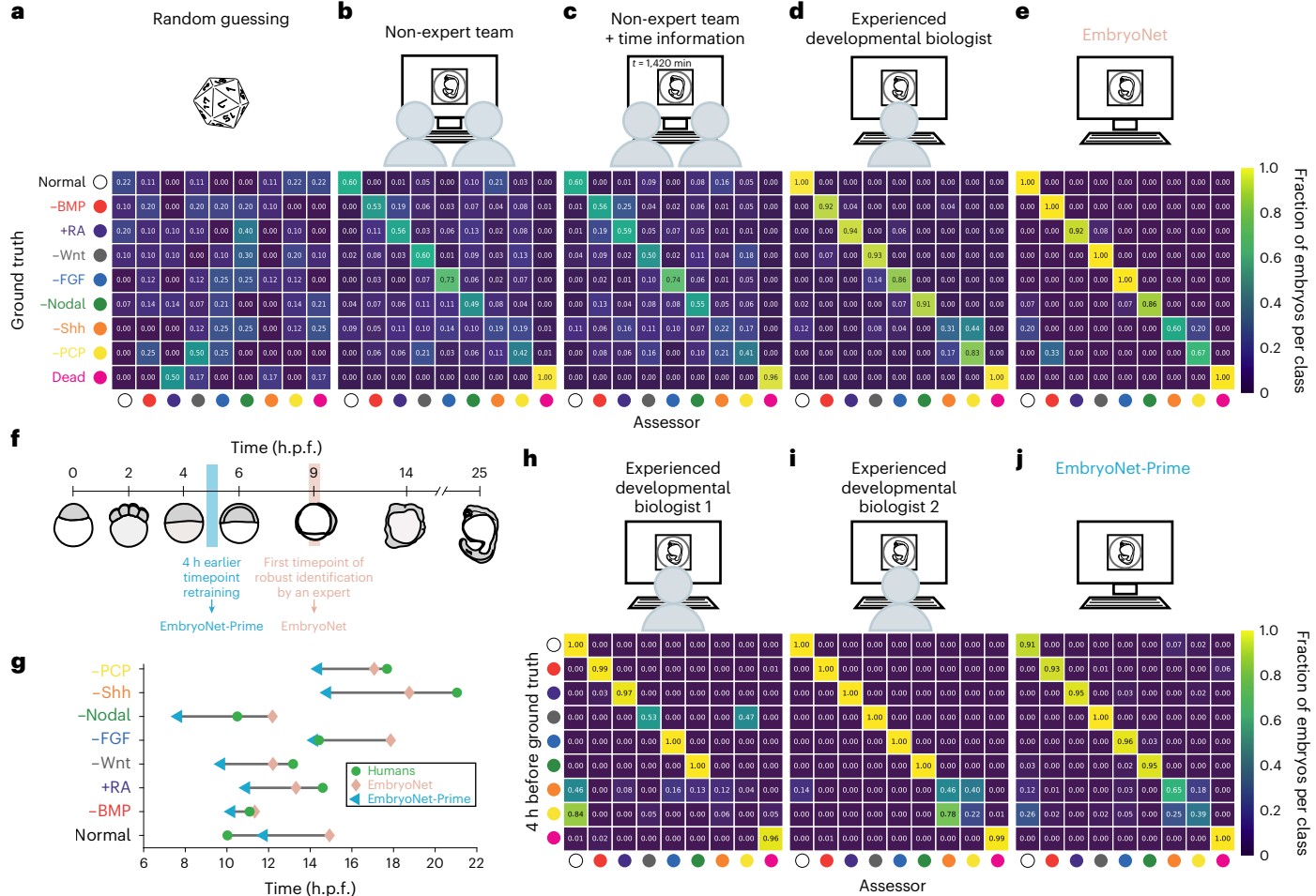

**Fig. 2 | Classification of 98 single embryo images by non-expert teams, experienced researchers and EmbryoNet. a–e**, Schematic set-ups and confusion matrices showing the classification of the respective labeler compared with the ground truth (human annotation, treatment known). Classification performance is shown as a heatmap and fractions of 1 for the classification of 98 single images by a pseudo-random number generator (**a**), by a non-expert team without (**b**) or with (**c**) additionally provided time information (average performance), by an experienced researcher (**d**) and by EmbryoNet (**e**). **f**, Schematic of embryo detection over time. To allow for earlier detection, we annotated the training data 4 h before (blue time frame) the timepoint at which they could be robustly

annotated by a labeler aware of the treatment (pink time frame). The embryo sketches show the phenotype of Nodal-inhibited samples at the respective time. The resulting network with earlier detection was termed 'EmbryoNet-Prime'. **g**, Characteristic times of detection for each class based on the assessment of human experts, EmbryoNet and EmbryoNet-Prime. $n_{Normal} = 74$, $n_{-BMP} = 119$, $n_{+RA} = 66$, $n_{-Wnt} = 70$, $n_{-FGF} = 74$, $n_{-Nodal} = 110$, $n_{-Shh} = 63$, $n_{-PCP} = 57$. **h–j**, Classification performance in the early detection of phenotypes. Confusion matrices show the classification of image series by the respective labeler compared with the ground truth (human annotation, treatment known; detection time shifted to 4 h earlier). The number of analyzed images is shown in Supplementary Tables 20–22.

identified embryonic phenotypes across classes with an overall accuracy of 79% (F-score = 0.78; Fig. 2d and Supplementary Tables 5 and 8). Strikingly, EmbryoNet outperformed both expert and non-expert human assessors on these images: it completed the task in a few seconds and with an overall accuracy of 91% (F-score = 0.90; Fig. 2e and Supplementary Tables 5 and 9), comparable to the performance across the entire validation dataset (see above).

To test whether context-dependent information could improve human classification performance, we asked two human experts to classify additional time-series experiments. Information about the specific treatment for each embryo was not disclosed to the assessors, but they were aware that all embryos in one video received the same treatment. Human classification performance slightly increased to an overall accuracy of 83% (F-score = 0.84). EmbryoNet still outperformed the human experts with an accuracy of 90% (F-score = 0.90), especially for the classification of the most difficult and subtle phenotypes (−Shh and −PCP) with F-scores of 0.54 (−PCP) and 0.72 (−Shh) compared with the human F-scores of 0.04 and 0.36, respectively (Extended Data Fig. 4a–d and Supplementary Tables 10–15). In addition, EmbryoNet

accurately identified phenotypes that were not fully penetrant, such as weaker BMP[14] and Nodal defects[46] (Extended Data Fig. 5 and Supplementary Tables 16–19).

Given EmbryoNet's performance in identifying subtle phenotypes, we hypothesized that we could leverage artificial intelligence to detect very early embryonic defects before they would be recognized by human experts. We therefore retrained EmbryoNet by moving the relevant developmental timepoint corresponding to each treatment class to 4 hours earlier, before the phenotype was obvious to a human annotator (Fig. 2f). Strikingly, the resulting network, EmbryoNet-Prime, was able to identify Nodal loss-of-function phenotypes at the beginning of gastrulation, several hours before human annotators could confidently recognize them (Extended Data Fig. 6), with an accuracy of 90% (F-score = 0.93; Fig. 2h–j, Extended Data Fig. 4e and Supplementary Tables 10 and 20–24). Similarly, the network detected the −BMP, +RA, −Wnt, −Shh and −PCP phenotypes on average 2–3 hours earlier (Fig. 2g and Extended Data Fig. 6), consistent with the known expression and activation profiles of the signaling molecules (Supplementary Note 1).

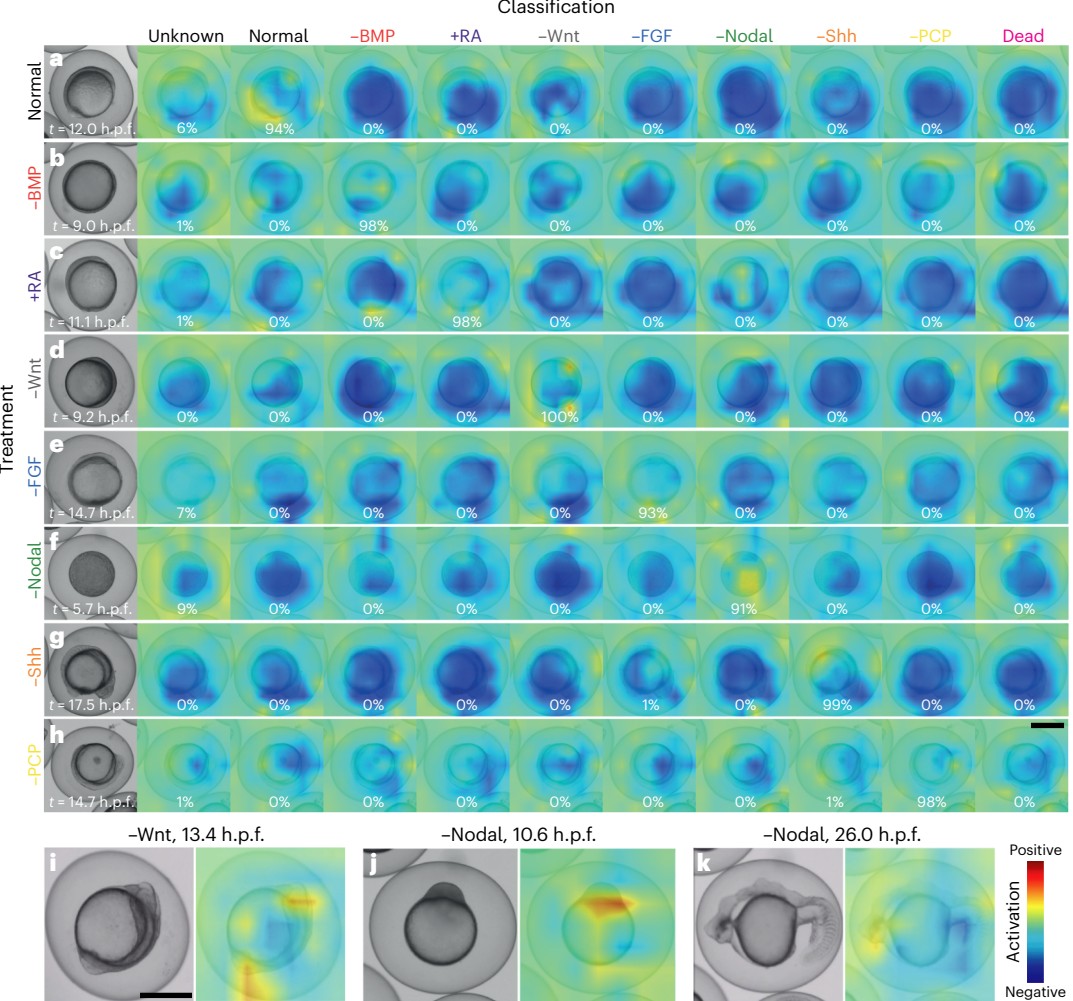

**Fig. 3 | Embryo features activating the neural network.** Class activation heatmaps based on the last convolutional layer of EmbryoNet-Prime showing the part of the image that activates the network at the given timepoint for normal (**a**) and signaling-defective embryos (**b**–**h**). Every signaling-defective embryo (−BMP (**b**), +RA (**c**), −Wnt (**d**), −FGF (**e**), −Nodal (**f**), −Shh (**g**), −PCP (**h**)) is displayed in all classification channels, but only the classification channels corresponding to the correct signaling manipulation show warm colors. The percentages represent the probability of detection. See Supplementary Note 3 for sample sizes. Also see Supplementary Videos 9–24. **i**–**k**, Selected embryos showing defects highlighted by the corresponding class activation heatmaps for −Wnt at 13.4 h.p.f. (**i**) and −Nodal at 10.6 h.p.f. (**j**) and 26 h.p.f. (**k**). Scale bars, 500 μm.

## EmbryoNet recognizes known and latent defect features

What could be the features that are detected by EmbryoNet-Prime, which facilitate earlier classification compared with human assessors? To address this question, we leveraged class activation map (CAM) visualization[47], which can be used to perform object localization without additional annotation by projecting the probability of the trained classes onto an input image. The resulting CAM should show the discriminative image regions used by the CNN to identify a class: positively activated domains should highlight image regions that support a particular class, whereas negative domains should show regions that oppose that class (Fig. 3 and Supplementary Note 3).

To evaluate the utility of CAM visualization, we first examined steep and sudden switches in classification. For example, BMP-inhibited embryos frequently disintegrate (Supplementary Video 2), switching from −BMP to Dead in terms of classification. Indeed, this classification switch can be observed in EmbryoNet-Prime's CAMs. Once −BMP embryos disintegrate, their CAMs in the −BMP channel immediately show negative activation accompanied by a positive activation in the CAMs for the Dead class (Supplementary Videos 11 and 12). These results indicate that CAM visualization can provide insight into the logic of phenotype classification.

Using this approach, we found that EmbryoNet-Prime identified known defects caused by the disruption of signaling pathways, but also detected latent features at an earlier developmental stage compared with human assessors. For example, Wnt mutants are known to exhibit prominent tail bud and head defects at 24 h.p.f.[11,12]. EmbryoNet-Prime was indeed activated in these regions at late stages (Fig. 3i). Strikingly, during early segmentation the whole body-axis already showed activation (Supplementary Videos 15 and 16), and the detection of head and tail defects also occurred as early as the bud stage (Fig. 3d, Extended Data Fig. 6 and Supplementary Videos 15 and 16). Thus, −Wnt embryos were detected earlier by EmbryoNet-Prime than by human assessors (Fig. 2g and Extended Data Fig. 6). Similarly, late-stage classification of −Nodal embryos by EmbryoNet-Prime relied on well-known defects in the ectodermal thickening (Fig. 3j), head, tail and trunk regions (Fig. 3k). Surprisingly, however, EmbryoNet-Prime was also able to classify early-stage −Nodal embryos (~6 h.p.f.; Supplementary Video 19) based on latent defects. The detection started with activation at the margin (Fig. 3f and Supplementary Videos 19 and 20) and continued with activation spots at the border between yolk and blastoderm, directly adjacent to the embryo proper. Although this fits well with known regions of Nodal expression and activity[37,48,49], it will

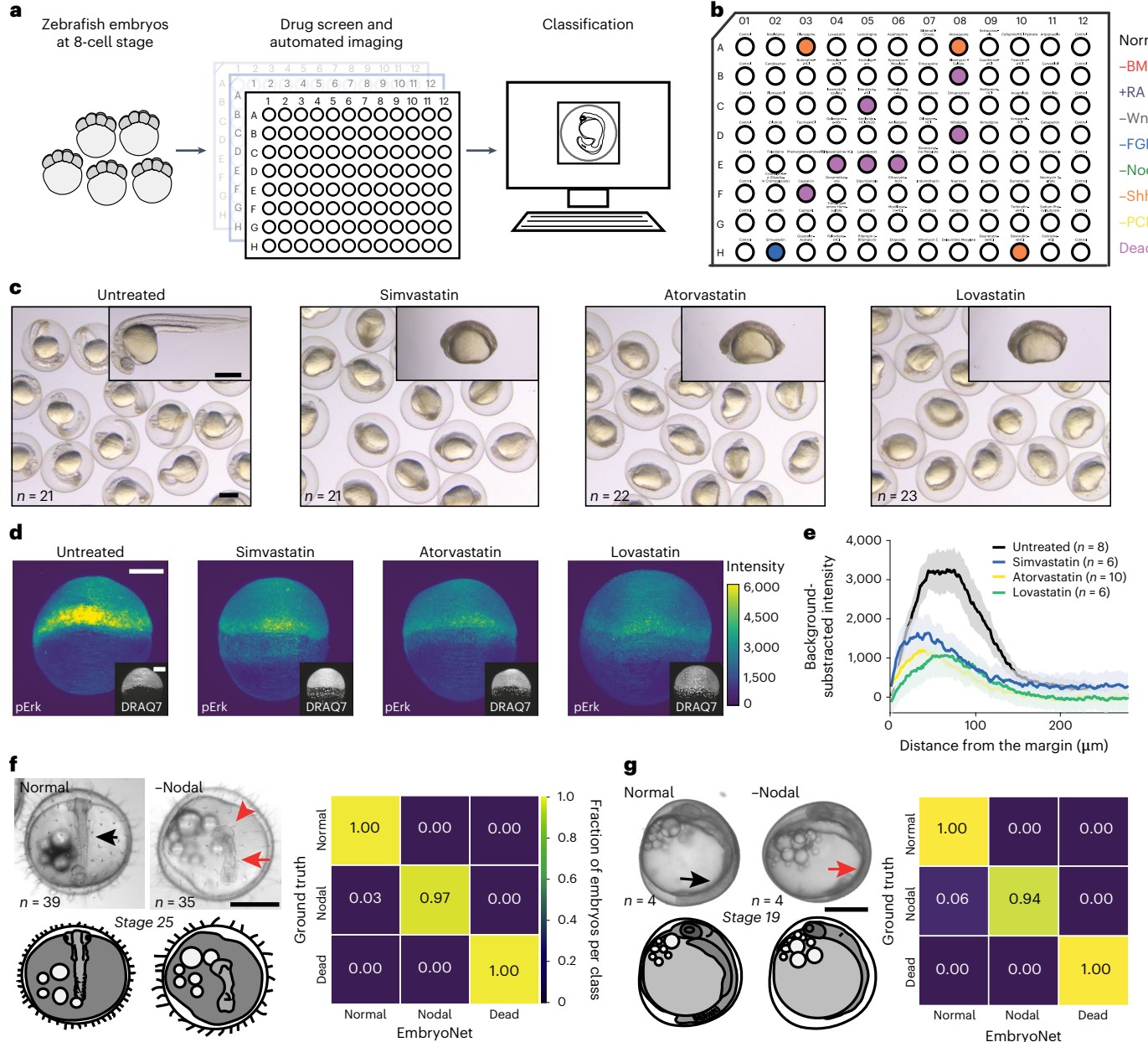

**Fig. 4 | Applications of EmbryoNet in drug screening and other species.**
**a,b**, Automated phenotype-based drug screening. **a**, Schematic of the phenomic drug screen. Embryos were exposed to compounds in 96-well plates and imaged for 24 h. Phenotypes were classified automatically by EmbryoNet. **b**, Layout of BML-2843 library plate 2 with majority phenotype classification for each well. Simvastatin in well H-02 was classified as −FGF. **c**, Statins identified by EmbryoNet in the drug screen caused body axis defects similar to −FGF loss-of-function phenotypes. **d**, Representative immunofluorescence images of the FGF signaling transducer pErk in untreated and statin-treated embryos, respectively. The representative images have pErk profiles that are closest to the mean signaling profile of each group. Images are shown at the same contrast and brightness. The inserted lower panels show cell nuclei labeled with DRAQ7. **e**, Quantification of background-subtracted pErk fluorescence intensity gradients in wild-type (black), simvastatin-treated (blue), atorvastatin-treated (yellow) and lovastatin-treated (green) embryos along the marginal-to-animal pole axis. The error envelopes show s.e.m. **f,g**, Extension of EmbryoNet to other species. **f**, Images of wild-type (left) and Nodal-deficient (right) medaka embryos with the confusion matrix of classification performance. **g**, Images of wild-type (left) and Nodal-deficient (right) three-spined stickleback embryos with the confusion matrix of classification performance. Black arrows point to somites in healthy embryos, while red arrows point out missing somites. The red arrowhead shows a mispatterned central nervous system. Scale bars: 500 μm (**c,f,g**) and 200 μm (**d**).

be interesting to determine how these molecular signatures manifest in latent cellular and morphological features.

**EmbryoNet can identify novel signaling modulators**

High-content image-based drug screens can be used to identify novel compounds affecting cellular phenotypes. However, large-scale drug screens assessing whole phenomes with rich biological information[50] are currently hampered by the need to visually assess a very large number of images, and are further complicated by the potential ambiguity of defects and variability between assessors. To determine whether EmbryoNet could be used to link chemical manipulations to signaling-based phenotypic defects, we performed a large-scale zebrafish screen using FDA-approved and bioactive compounds (Fig. 4a).

We screened approximately 1,000 compounds using 96-well plates containing four to five zebrafish embryos per well. The screen spanned 2–26 h.p.f. We first tested well-known viability modulators with characterized mechanisms of action such as aphidicolin, bafilomycin A1, blebbistatin, brefeldin A, cycloheximide, cytochalasin B, latrunculin B, staurosporin, trichostatin A, tunicamycin and vinblastine. EmbryoNet reliably classified embryos treated with these substances as Dead, while classifying mock-treated embryos as Normal (Supplementary Tables 25–35, Extended Data Figs. 7 and 8 and Supplementary Videos 25–35). EmbryoNet also correctly detected known modulators of signaling pathways, such as the RA agonists all-trans-retinoic acid and TTNPB (Extended Data Fig. 7 and Supplementary Table 26).

Importantly, for some small molecules we identified previously unrecognized effects on signaling pathways in embryos. This group includes several hydroxymethylglutaryl-coenzyme A reductase inhibitors, a class of compounds also known as statins[51–54]. Interestingly, embryos treated with several statins, including simvastatin, atorvastatin and lovastatin, were classified as −FGF by EmbryoNet (Fig. 4b, Supplementary Videos 36–38 and Supplementary Tables 29, 32 and 35). Consistent with the −FGF classification, embryos treated with these drugs showed defects in dorsal–ventral patterning[51,52] and had loss of posterior tissues typical of −FGF embryos[13,55–57] (Fig. 4c). Strikingly, the activity of the FGF signal transducer pErk was also reduced in statin-treated compared with untreated embryos (Fig. 4d,e), possibly due to dampened isoprenylation of the upstream small GTPase Ras[58]. According to patient information regarding side-effects and databases of potentially embryotoxic teratogenic therapeutics, the intake of selected hydroxymethylglutaryl-coenzyme A reductase inhibitors such as lovastatin is not recommended during pregnancy and lactation (Supplementary Note 4). Notably, simvastatin is recommended as a substitute, although EmbryoNet detected the same defects in response to this drug. However, the bioavailability in zebrafish compared with human cells is currently unclear.

In summary, our drug screen shows that EmbryoNet can be used to identify teratogenic effects caused by bioactive compounds and to associate them with signaling pathways.

### Generalization of EmbryoNet to other species

To test the generalizability of our approach, we next applied EmbryoNet to identify signaling defects in the evolutionarily distant species medaka (*Oryzias latipes*) and three-spined stickleback (*Gasterosteus aculeatus*). These fish diverged from zebrafish hundreds of millions of years ago[59,60]. We adjusted the imaging length of our recordings to match the slower developmental speed of both species[61,62] and modified species-specific parameters such as temperature, number of embryos per well and drug concentration as needed.

We found that in both medaka and stickleback, wild-type animals had well-formed somites (Fig. 4f,g, black arrows) and eyes (Supplementary Videos 39 and 41), while Nodal-inhibited embryos showed a loss of somites (Fig. 4f,g, red arrows) concomitant with severe central nervous system defects and frequent cyclopia (Fig. 4f, red arrowhead; Supplementary Videos 40 and 42). After training with these datasets, EmbryoNet robustly identified wild-type and Nodal-inhibited individuals in both species (Fig. 4f,g). These results support the broad applicability of EmbryoNet in identifying signaling-based complex phenotypic defects in different species.

### Discussion

Phenome refers to the entire set of phenotypes of an organism over time, and phenomics has emerged as a promising approach for connecting these phenotypes with the underlying genotypes and environmental influences[50]. A quantitative understanding of how the phenome changes in response to genetic mutations and environmental stimuli would be highly informative, but phenomics requires the processing of large amounts of high-dimensional data[36,63–66]. Computer vision and machine learning techniques are therefore promising approaches for advancing this field and indeed are increasingly being applied in plant and crop phenomics[67]. Here, we present a machine learning-assisted method for the robust phenomic analysis of developmental defects during vertebrate embryogenesis.

The automated phenotyping tool that we developed, EmbryoNet, is based on CNNs. Strikingly, EmbryoNet outcompeted human assessors in terms of speed, accuracy and sensitivity. Assessing zebrafish embryos, EmbryoNet was able to quickly and accurately link phenotypes to major signaling pathways, including classifying incompletely penetrant phenotypes. We were also able to retrain EmbryoNet for assessing other fish species separated from zebrafish by hundreds of millions of years in evolution, enabling the analysis of high-dimensional phenomic data in different taxa. EmbryoNet may thus be able to accelerate the characterization of developmental mutants in multiple species. Finally, in a proof-of-concept drug screen with two drug libraries, we showed that EmbryoNet correctly associated compounds with signaling functions. We therefore believe that this approach can be used to understand the signaling effects of various compounds and medications, thus opening up the possibility of applying drugs to new therapeutic contexts and applications.

While EmbryoNet offers significant advantages in identifying phenotypes at earlier developmental stages, there are some caveats and weaknesses to consider. It remains uncertain whether EmbryoNet can outperform humans in detecting very mild phenotypes, such as those caused by low drug concentrations. Additionally, its reliance on a library of manual annotations limits its ability to classify novel phenotypes, particularly those arising from the combinatorial disruption of signaling pathways. The rapid development of deep learning technologies could be leveraged to enhance EmbryoNet's capabilities and help address EmbryoNet's current limitations. By building on these technological breakthroughs, it may become possible in the future to bridge the genotype–phenotype gap and tackle the long-standing question of how diverse body plans are genetically encoded[68].

We provide EmbryoNet as open-source software, with Python packages, a GitHub repository and GUIs for labeling data and phenotype classification (http://github.com/mueller-lab/EmbryoNet). We also provide the training, testing and the drug screen imaging data as a resource to the community (http://embryonet.uni-konstanz.de, http://github.com/mueller-lab/EmbryoNet). Due to its modular open-source nature, EmbryoNet can be easily adapted to a variety of purposes, including embryos of other species and organoids, in which automated phenotyping will expedite biological and pharmaceutical discovery.

### Online content

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

## Methods

### Embryo preparation

The experiments were performed exclusively with embryos and larvae that were not yet freely feeding. All procedures and zebrafish, medaka and stickleback husbandry were carried out in accordance with the guidelines of the European Union directive 2010/63/EU and the German Animal Welfare Act as approved by the local authorities represented by the Regierungspräsidium Tübingen and the Regierungspräsidium Freiburg (Baden-Württemberg, Germany).

Zebrafish embryos of the TE strain were collected from batch crosses within 1 h after fertilization. Fertilized embryos were manually selected using a glass Pasteur pipette. At this timepoint, zebrafish embryos were between the 2- and 8-cell stages. A total of 10–20 embryos were pipetted into each well of a 24- or 48-well plate in 1 ml zebrafish embryo medium[48]. Small-molecule signaling pathway agonists and antagonists were added by pipetting them into the filled well with the final concentrations listed in Supplementary Table 1. To obtain PCP phenotypes, 1 ng *vangl2*-targeting morpholino[69] was injected at the one-cell stage. Shh treatment was carried out either by cyclopamine incubation or *gli3R-GFP* mRNA injection. For overexpression of signaling antagonists, 10 pg *lefty1* mRNA[48] together with 0.1 ng of 10 kDa Alexa647-coupled dextran (Invitrogen D22914) or 75 pg *chordin* mRNA[70] together with 0.1 ng of 10 kDa Alexa488-coupled dextran (Invitrogen D22910) were injected into one-cell stage wild-type zebrafish embryos. To validate phenotypes, *swirl* homozygous mutants[70] and maternal zygotic homozygous *oep* mutants[41] were used. All zebrafish embryos were between 0 h.p.f. and 48 h.p.f.

Medaka eggs of the Cab strain were collected from standard crosses and separated with forceps. They were incubated at 28 °C in medaka embryo medium (17 mM NaCl, 0.4 mM KCl, 0.27 mM CaCl$_2$, 0.65 mM MgSO$_4$) and distributed into 24-well plates. Small-molecule signaling pathway antagonists were added at the early blastula stage to a concentration of 7.5 µM. Embryos were imaged from 8 h.p.f. until 45 h.p.f. with intervals of 5 min at 28 °C. Approximately 500 Medaka embryos between 0 h.p.f. and 48 h.p.f. were used for training and testing.

Stickleback embryos of wild-derived marine strains from Little Campbell River (Canada) and Tyne River (Scotland) were obtained by in vitro fertilization and incubated until 20 h.p.f. at 16 °C in stickleback embryo medium (3.5 g l$^{-1}$ instant ocean salt in reverse osmosis water). The eggs were separated using brushes and distributed into 48-well dishes. The small-molecule signaling pathway antagonists were applied at a concentration of 15 µM, and embryos were imaged for 120 h with intervals of 5 min at 15–18 °C. Approximately 200 stickleback embryos between 0 h.p.f. and 140 h.p.f. were used.

### Image acquisition

Images were acquired using an ACQUIFER Imaging Machine (DITABIS AG) with a white light-emitting diode (LED) for bright-field imaging and a scientific complementary metal oxide semiconductor 2,048 × 2,048 camera (Hamamatsu sCMOS 2k × 2k) in a single plane with a ×2 Plan UW numerical aperture 0.06 objective (Nikon) using the Imaging Machine software (v4.00.21). The integration time was fixed at a 110 ms exposure time and 100% relative LED intensity in the bright-field channel. Imaging was performed at 28 °C with 720 iterations at intervals of 120 s. Images were stored as 12-bit TIFF files at a size of 2,048 × 2,048 pixels and 0.31 pixels µm$^{-1}$ and converted to JPEG or PNG files for further phenotype analyses.

To generalize the method independently of the microscope, a Keyence BZ-X810 microscope equipped with a ×2 apochromat objective, a 3.7 W LED lightsource and the BZ-X800 viewer software (Keyence, v01.03.00.01) was also used to acquire embryo images (Supplementary Tables 2, 14, 15, 23 and 24 and Extended Data Fig. 4d,e). The exposure time was 0.1 ms with 50% relative intensity. The images were stored as 8-bit JPEG files at a size of 1,920 × 1,440 pixels.

Medaka and stickleback eggs contain large lipid droplets and, compared to zebrafish, have a larger yolk in relation to the embryo proper. Additionally, medaka eggs are surrounded by adhesive filaments. Given that these features are visually very prominent, the embryos were required to be imaged until the late segmentation stages to robustly detect morphological differences.

### Embryo detection

A dataset of manually annotated embryos was generated using the GUI FishLabeler (http://github.com/mueller-lab/EmbryoNet). The dataset was split into two subsets: 90% of the images were used for training and 10% for validation. Additionally, an independent manually annotated dataset was generated for testing.

Individual embryos were automatically detected at each image frame of the acquired movies using a standard object-detection algorithm based on the Hough transform[71]. The location of individual embryos was computationally determined using bounding boxes. The range of embryo radii in pixels was provided according to the microscope acquisition parameters for each experiment independently. As output, a set of JSON files containing the information about the bounding boxes of individual embryos was generated. The Hough transform-based embryo detector can be replaced by other object recognition methods (such as watershed segmentation) to detect non-spherical species (for example, *Drosophila melanogaster*).

### Embryo tracking

To obtain information about the whole developmental path of each individual, the embryos identified at individual frames were grouped using an object-tracking approach. Detections of the same embryo in consecutive frames were confirmed using the DeepSort algorithm without re-identification[72].

### Manual labeling of training datasets

All embryos were initially set as class Unknown. Then, each embryo track was manually annotated with its specific phenotypic class (that is, Normal = Wild type, −BMP, +RA, −Wnt, −FGF, −Nodal, −Shh, −PCP) from the timepoint when the phenotype could be observed by an experienced annotator. Additionally, embryos that disintegrated were labeled as Dead. Embryos that were only partially in the image or that showed an unspecific phenotype were annotated as Cut and were excluded from the training, validation and test datasets. Additionally, the −BMP and −Nodal classes were subclassified into severity levels: weak, ~30% phenotype severity; intermediate, ~60% phenotype severity; and severe, ~100% phenotype severity. For Nodal phenotypes, the percentage bins were determined by the concentration of the inhibitor, with 100% corresponding to the minimum concentration that led to full penetrance. The drug concentration was then directly used as a binned fraction of the fully penetrant dose as ground truth for the severity bins, accepting a certain spread of phenotypes. Binning of BMP severity levels was done based on previous classification schemes[14], with the class C3 corresponding to 30%, C4 to 60%, and C5 to 100% severity. Altogether 14 classes were obtained for the classification process. The annotator had previous knowledge about the treatment of the respective embryo, and the expertise to recognize all of the phenotypes.

To train EmbryoNet-Prime for an earlier detection, the original manual annotations were used to determine the timepoint when the majority of the embryos were classified correctly. The appropriate class was then assigned 120 frames (4 h) earlier, when the phenotype could not be identified by eye. Cut and Dead embryos were not changed.

### Model training and embryo classification

The use of embryo images as the only input could lead to misclassifications between embryo phenotypes, which have a similar appearance at different developmental stages. To increase classification performance, the developmental timepoint was added as a second input to

the classification algorithm. In total, four channels were used as input for model training. The first three channels correspond to a standard RGB image, and the remaining one is a 'timestamp' channel representing the time that has passed from the beginning of the experiment. The size of the images was 224 × 224 pixels. The timestamp was linearly mapped from the real developmental time to the domain [0, 1], where 0 corresponds to ~2 h.p.f. and 1 to ~26 h.p.f. Given that the input classes were imbalanced (Supplementary Table 2), the overrepresented class Unknown was undersampled and the 13 remaining classes replicated[73].

For the classification task, a modified version of the widely used ResNet18[40] architecture was selected. The network architecture was chosen due to its easy and fast convergence in image classification tasks. The ResNet18 model was modified by using a time channel as additional input and thus feeding four instead of three channels, and by replacing the last classification layer with the current classification layer. Time was also used as input to the last fully connected layer. The parameters of the neural network weights, unlike the neural network architecture (that is, the mathematical function structure describing an artificial neural network), were changed during the training procedure via a back-propagation algorithm.

The CNN model was trained using the supervised back-propagation training method[74], a common algorithm for training neural networks. The Adam optimizer was used, which is a back propagation-based optimization algorithm that determines the value change of neural network weights based on the loss function gradient. Softmax cross-entropy was used as the loss function (that is, the penalty for a poor prediction, indicating how bad the model's prediction was on a single example):

$$L = -1/n \sum ln \, p_i$$

where $p_i$ is the index of the correct probability of the $i$-th image, and $n$ is the number of images in the batch. In the case in which the model's prediction is perfect, the loss would be zero; otherwise, the loss is greater. Cross-entropy loss is a common loss function used in machine learning and it measures the expected negative logarithm's value for the correct classification probability.

The Albumentations library[75] was used to increase the amount of training images by adding slightly modified copies of the existing images. Augmentations were applied during the training process, including random horizontal and vertical flips, rotations in the range of 1–90° with steps of 1°, crops and salt-and-pepper noise (Supplementary Table 3). During the training process a random augmentation from each group was picked and applied to the input image.

Given that the selected CNN model did not converge when all datasets and augmentations were used from the beginning of the training, a progressive training design involving different levels of difficulty was developed. In brief, the training was performed sequentially by dividing it into several steps with progressive addition of data and augmentations. At each step extra data were added as input and new augmentations applied at each epoch, that is, at each pass over the entire training dataset during the training procedure. The initial learning rate was set to $10^{-3}$ and it decayed by a factor of 0.1 after each epoch. The learning rate, that is, the parameter by which the loss gradient value is multiplied during each iteration, was restored to the value of $10^{-3}$ at the beginning of each iteration. The model was trained using eight steps with 10–20 epochs per step, resulting in a total of 152 epochs. For the whole training, a batch size (that is, the number of training examples used in one iteration) of $n = 350$ was used, and the training was performed on an NVIDIA RTX 3090 card in Ubuntu 20.04.4 LTS.

Rotation- and mirror-invariance of the embryo appearance was exploited to boost the classification performance by running the trained network for each detected embryo eight times: once with the original embryo image, once with the image flipped horizontally, then flipped vertically, then mirrored diagonally, and then with each of these samples rotated by 90°. Following this step, the classification probabilities were averaged. Each embryo was assigned to the class that had the maximum probability.

## Model transition logic

To further improve the results of the classification, the information from embryo tracking as well as previous knowledge about transitions between phenotypes was incorporated into the classification task. In brief, first the classification results of the CNN for each embryo track were collected and transitions between classes identified. The only biologically possible transitions in an embryo track were set as follows: from Unknown to a phenotype class, from a phenotype class to Dead, or from Unknown to Dead. Any other transition in an embryo track was penalized in the model prediction. The quality of the whole track model prediction was evaluated by computing the number of frames between transition points with the class expected by the model being analyzed. The transition sequence that achieved the least cost was considered to be the correct one. The outliers were then ignored in the track history. Nodal and BMP severity classifications were similarly corrected by selecting the severity class that was most frequently observed over the timecourse.

For medaka, a semi-supervised training method was used by assigning a classification transition point from which the phenotypes were easy to distinguish for a human and automatically applying this to all training data. Given that medaka embryos disintegrated if they were treated with Nodal inhibitor before the blastula stages, the medaka experiments did not start at cleavage but at blastula stages. This opportunity was used to set the transition point to timepoint 1, such that the Unknown class did not have to be used at all. This did not reduce the training or classification efficiency (Fig. 4f), showing that an Unknown state for early stages is not unconditionally required.

## Evaluation of classification efficiency

For the performance measure of classification, subset accuracy was computed. Subset accuracy is the fraction of images $n$ that were classified correctly:

$$\text{Accuracy} = \frac{1}{n} \sum_{i=1}^{n} I(\check{y}_i = y_i)$$

F-scores were calculated as

$$\text{F} - \text{score} = 2 \times \frac{\text{Precision} \times \text{Recall}}{\text{Precision} + \text{Recall}}$$

with

$$\text{precision} = \frac{\text{true positives}}{\text{true positives+false positives}} \text{ and}$$

$$\text{recall} = \frac{\text{true positives}}{\text{true positives+false negatives}}$$

In the confusion matrices, the class Unknown was not taken into account. The numerical data for the confusion matrices including the class Unknown are provided in Supplementary Tables 4, 9, 11–15 and 17–24, and this class was also included in the overall metrics of accuracy and F-score.

The evaluation dataset for Fig. 2a–e was generated by compiling three stacks of 98 images each selected from the full test dataset (Fig. 2h–j and Extended Data Fig. 4). To evaluate the performance of random guessing, the function randi from MATLAB R2022a was used for the generation of a pseudo-random scalar integer between 1 and $n_c$, where $n_c$ is the number of classes. The image stacks were then labeled with the classes corresponding to the pseudo-random numbers and evaluated for performance by calculating accuracy and F-score. The non-expert teams received one randomly selected image stack for their assessment task. The experienced developmental biologist assessed all three image stacks, and average performance is shown in Fig. 2d.

## CAMs

To visualize the regions of images that influenced the model to make classification decisions, CAMs were used. To visualize the CAMs generated by EmbryoNet, the weights of the final output layer in a fully connected layer were projected using global max pooling, as previously proposed[47]. This approach enabled the visualization of regions positively or negatively activated for a particular class. CAMs were calculated for all classes, and their values were normalized so that the minimum and maximum values for all classes correspond to −1 and +1, respectively. To improve the visualization of areas with large positive or negative values (that is, relevant regions for the decision), the CAMs were remapped using the following function:

$$\bar{V}_{CAM} = \text{sgn}(V_{CAM}) \times \sqrt{V_{CAM}}$$

where $V_{CAM}$ are the normalized values of the CAMs, $\text{sgn}(\cdot)$ is the sign function and $\bar{V}_{CAM}$ are the remapped CAM values. Finally, the values of the CAMs were mapped to 8-bit and visualized with the jet colormap.

## Drug screening

Plates of the Screen-Well ICCB Known Bioactives Library BML-2840-0100 and the FDA-approved drug library BML-2843 were defrosted at 22 °C for 1 h and centrifuged at 1,890 ×g for 2 min (Eppendorf 5810 R). Six 96-well microtiter plates were pre-filled with 96 µl cell culture-grade PBS (Gibco). From each library plate, 4 µl per well were transferred, resulting in a 1:25 dilution. Blank wells were filled with 4 µl cell culture-grade PBS. Zebrafish embryos were collected as described above, but selected embryos were washed three times with 200 ml embryo medium and transferred to a 96-well plate (Greiner Bio-One), three to five embryos per well. Each well was filled with embryo medium to a volume of 135 µl. Subsequently, 15 µl solution were transferred from the 1:25 intermediate dilution plates to each well containing embryos. Plates were covered with transparent foil, and a plastic lid was placed on the plate.

Screening plates were placed in the ACQUIFER Imaging Machine as described above with an imaging interval between 135 s and 192 s. Image files were converted to JPEG files for further phenotype analyses. The images from the 96-well screening plates were sorted into separate directories related to respective wells using a custom Python script (Drug screen script 1; http://github.com/mueller-lab/EmbryoNet/tree/main/Train_Eval/tools/DrugScreen). The data files were read into the custom FishClassifier software and evaluated for detected phenotypes. For each image file, phenotype detections were stored as a separate JSON file. The JSON files were read using a custom Python script (Drug screen script 2; http://github.com/mueller-lab/EmbryoNet/tree/main/Train_Eval/tools/DrugScreen). Evaluated phenotypes were linked with corresponding treatments and finally stored as Excel files, containing the number of images for each class in the time series. These files were used to generate charts for predicted phenotypes resulting from each treatment (Drug screen script 3; http://github.com/mueller-lab/EmbryoNet/tree/main/Train_Eval/tools/DrugScreen). The majority phenotype for each well was determined as the class to which the highest number of embryo images was assigned.

## Retest and characterization of statins in FGF signaling

Zebrafish embryos were treated with 20 µM simvastatin in embryo medium (Enzo Life Science BML-G244-0050, final concentration of DMSO solvent: 0.2%), 40 µM atorvastatin (Sigma PHR1422, final concentration of DMSO solvent: 0.4%) or 0.4 µM lovastatin (PHR1285, final concentration of DMSO solvent: 0.04%) starting at 1.5–2 h.p.f. or were left untreated and incubated at 28 °C.

Live embryos were imaged at 28 h.p.f. with a bright-field microscope (Leica M205 FCA). For close-up images, embryos were manually dechorionated using precision forceps and embedded in 2% methylcellulose in embryo medium.

For pErk immunostainings, untreated and statin-treated embryos were fixed at the shield stage with 4% formaldehyde in PBS overnight at 4 °C and then stepwise (25%, 50%, 75% methanol in PBST (PBS containing 0.1% Tween-20)) dehydrated. After an overnight incubation in 100% methanol at −20 °C, embryos were rehydrated in three steps (75%, 50%, 25% methanol in PBST). After permeabilization with ice-cold acetone for 20 min at −20 °C and additional washing steps with PBST, samples were blocked in 10% FBS in PBST for 2 h and incubated in 1:5,000 mouse anti-pERK antibody (Sigma, M8159) in 10% FBS in PBST overnight at 4 °C. Embryos were then washed at least 12 times with PBST, followed by another blocking step for 2 h with 10% FBS in PBST and overnight incubation with 1:5,000 donkey anti-mouse HRP-coupled secondary antibody (Jackson ImmunoResearch, 715-035-150) in 10% FBS in PBST at 4 °C. After washing at least 12 times with PBST and once with TSA 1x amplification buffer, embryos were incubated in 75 µl 1:75 Cy3-TSA in 1x amplification buffer for 45 min, protected from light. After washing for at least four times with PBST, embryos were incubated in 0.3 µM DRAQ7 (Invitrogen, D15106) in PBST for 30 min and then washed at least three times with PBST. Before imaging, stained embryos were wrapped in aluminum foil and stored overnight at 4 °C.

Fixed and stained embryos were mounted in 1.5% low-melting point agarose (Lonza, 50080) using a glass capillary (50 µl, Brand 701908) and imaged with a ZEISS Lightsheet Z.1 microscope using ZEN 3.1 Black Edition acquisition software. The imaging chamber was filled with water, and filters and lightsheets were auto-aligned prior to imaging[76]. Embryos were positioned with the brightest pErk signal pointing towards the imaging objective (presumptive dorsal view). For each embryo, z-slices with 5 µm between each slice were acquired. All images were acquired with dual lightsheet illumination using a W Plan-Apochromat ×10 objective at ×0.9 zoom, with laser powers of 2% and 6% for pErk and nuclei, respectively.

To measure spatial intensity profiles from the margin to the animal pole, maximum intensity projections of 75 z-slices were generated using Fiji[77], and pErk intensity profiles were calculated as follows. First, a rectangular region of interest with a width of 300 pixels was manually drawn from the margin of the blastoderm to the animal pole. Only images of embryos that were oriented with the dorsal side facing the camera were used for the analysis. The dorsal side could be identified after generating maximum intensity projections from image stacks. Embryos with tilted dorso-ventral axes were excluded. Then, the average intensity along the profile was calculated using the function Measure in Fiji. The background intensity of pErk was estimated as the median intensity value of the profiles of untreated embryos at the animal pole (between 250 µm and 280 µm from the margin) and subtracted from the intensity profiles using MATLAB 2022a (https://doi.org/10.48606/55).

## Reporting summary

Further information on research design is available in the Nature Portfolio Reporting Summary linked to this article.

## Data availability

Training and evaluation datasets for EmbryoNet are available from http://embryonet.uni-konstanz.de and https://doi.org/10.48606/15. The drug screen data are available from https://doi.org/10.48606/37, https://doi.org/10.48606/38 and https://doi.org/10.48606/41. Additional data that support the findings of this study are available from https://doi.org/10.48606/53 and https://doi.org/10.48606/55.

## Code availability

The source code for EmbryoNet is available from http://github.com/mueller-lab/EmbryoNet (https://doi.org/10.5281/zenodo.7531593). Additional custom scripts used for data analysis in this study are available from https://doi.org/10.48606/15.

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

## Acknowledgements

This project has received funding from the European Research Council (ERC) under the European Union's Horizon 2020 research and innovation program (grant agreement No. 863952 (ACE-OF-SPACE) to P.M.). This work was also funded by the EMBO Young Investigator Programme (P.M.), Max Planck Society (P.M., F.J.), the FWF (Project J-4507, D.Č.), the IZKF of the Medical Faculty of the University of Tübingen (P.M., N.T.), and the Deutsche Forschungsgemeinschaft (DFG, German Research Foundation) under Germany's Excellence Strategy— EXC 2117—422037984 (P.M.). We thank K.W. Rogers, A. Schauer, K. Sarieva, I. Carmi and M. Rössler for scientific input, consulting and illustration support. We also thank O. Aust for initial drug screenings and A. Baccini for technical assistance with compound retesting. For participating in the classification of single embryo images, we thank M. Akyüz, L. Amann, A. Balb, A. Bangnowski, S. Baumgärtner, A.-S. Becker, S. Berber, S. Bergemann, T. Berger, L. Betz-Jung, L. Beuten, M. Brückner, L. Budig, N. Bürgers, P. Buslaps, D. Casaburi, L. Dangel, J. Davia, T. Decker, A. Eiberle, J. Engler, C. Feldmann, M. Franz, E. Frese, A. Fronius, B. Goldschmidt, C. Gomes, D. Gaßebner, L. Haas, L. Haßfeld, L. Heger, L. Helten, S. Hillman, S. Hinte, L. Huber, J. Iffelsberger, I. Jorzik, J. Jung, L. Kammerer, J. Klein, E. Kleinke, H. Klenk, V. Kneipp, M. Kölle, M. Kröner, V. Kuhn, P. Kukofka, J. Küpfer, Y. Lan, K. Land, C. Lewin, M. Lohmer, J. Lüders, X. Lyu, H. Mahl, R. Manukjan, M. Martini, A. Maslonka, P. Matijas-Graf, N. Meier, T. Morell, F. Natale, M. Nyesö, F. Piehler, A. Pirker, S. Rampp, V. Raupp, K.M. Reagan, A. Reiß, G. Rösler, F. Roßmann, J. Roylands, L. Ruf, J. Schiele, R. Schmidt, A. Schneider, M. Schön, M. Schröter, A. Schupp, F. Stiller, S. Stöckl, L. Thellmann, M. Thomann, D. Torcuk, Z. Umbach, R. Unsöld, C. Vogl, H.C. von Vegesack, R. Wagner, G. Wallig, M.A. Wannemacher, L. Wanner, F. Welsch, C. Wolfer, V. Zickenberg and M. Ziefle. We also thank T. Thumberger and J. Wittbrodt for providing the medaka Cab strain, and P. Huang for providing the *pCS2-zGli3R-EGFP* plasmid.

## Author contributions

P.M. and B.J. conceived the study. P.M. supervised the project. D.Č., J.B., M.S., N.T. and H.M.-N. manually annotated images. M.S. and G.A. developed the software for EmbryoNet. M.S., G.A. and H.M.-N. developed software for downstream analysis. N.T. and D.Č. performed the drug screens. N.T. analyzed the drug screen. H.M.-N. analyzed the zebrafish data, M.S. analyzed the medaka data and N.T. analyzed the stickleback data. A.K. performed the retests of the statins, carried out immunostainings and lightsheet microscopy, and acquired time series on the Keyence BZ-X810 microscope. H.M.-N. contributed to lightsheet imaging and performed pErk quantification. D.Č. carried out all other experiments. J.H., S.K. and F.J. provided in vitro fertilized stickleback embryos. D.Č. and P.M. wrote the paper with input from all of the authors.

## Competing interests

The authors declare no competing interests.

## Additional information

**Extended data** are available for this paper at https://doi.org/10.1038/s41592-023-01873-4.

**Correspondence and requests for materials** should be addressed to Patrick Müller.

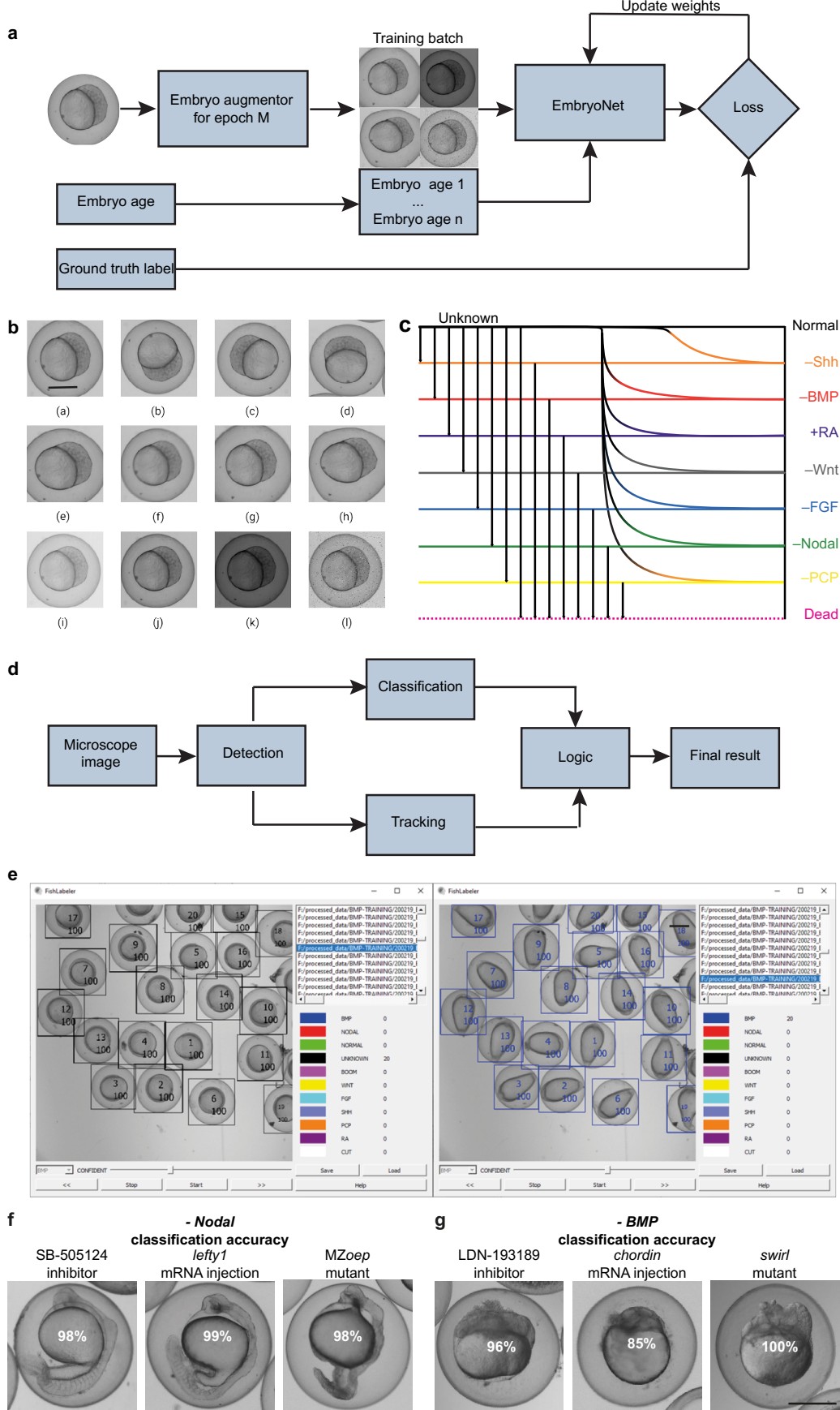

**Extended Data Fig. 1 | See next page for caption.**

**Extended Data Fig. 1 | Schematic of the training pipeline for EmbryoNet.**
**(a)** Overview of a training iteration. Augmented embryos were collected into a training batch with known embryo age. After EmbryoNet processed a batch of input images with ages, network outputs were compared with ground truth values. Based on cross-entropy loss, EmbryoNet weights were updated to minimize the loss. **(b)** Examples of augmentations used. **(c)** In our model transition logic, embryos have a limited set of allowed class transitions. All start in the class Unknown and can transition to any other class, from where they can go only to Dead, but not to other classes. Other transitions were assigned a cost. The model with the least cost was selected. **(d)** Schematic of the classification pipeline. **(e)** Graphical user interface (GUI) of EmbryoNet.

**(f,g)** Comparison of EmbryoNet's performance to recognize phenotypes induced by signaling modulation using small-molecule inhibitors, overexpression of signaling antagonists or pathway mutants. Nodal phenotypes (f) induced by small-molecule inhibitor treatment (SB-505124, n=33), injection of a pathway antagonist (*lefty1* mRNA, n=27) or in a receptor mutant (*MZoep*, n=27) were all classified by EmbryoNet as −Nodal with similar accuracy. BMP phenotypes (g) induced by small-molecule inhibitor treatment (LDN-193189, n=45), pathway antagonist injection (*chordin* mRNA, n=26) or in a pathway ligand mutant (*swirl$^{-/-}$*, n=13) were all classified by EmbryoNet as −BMP with similar accuracy. Scale bars: 500 μm.

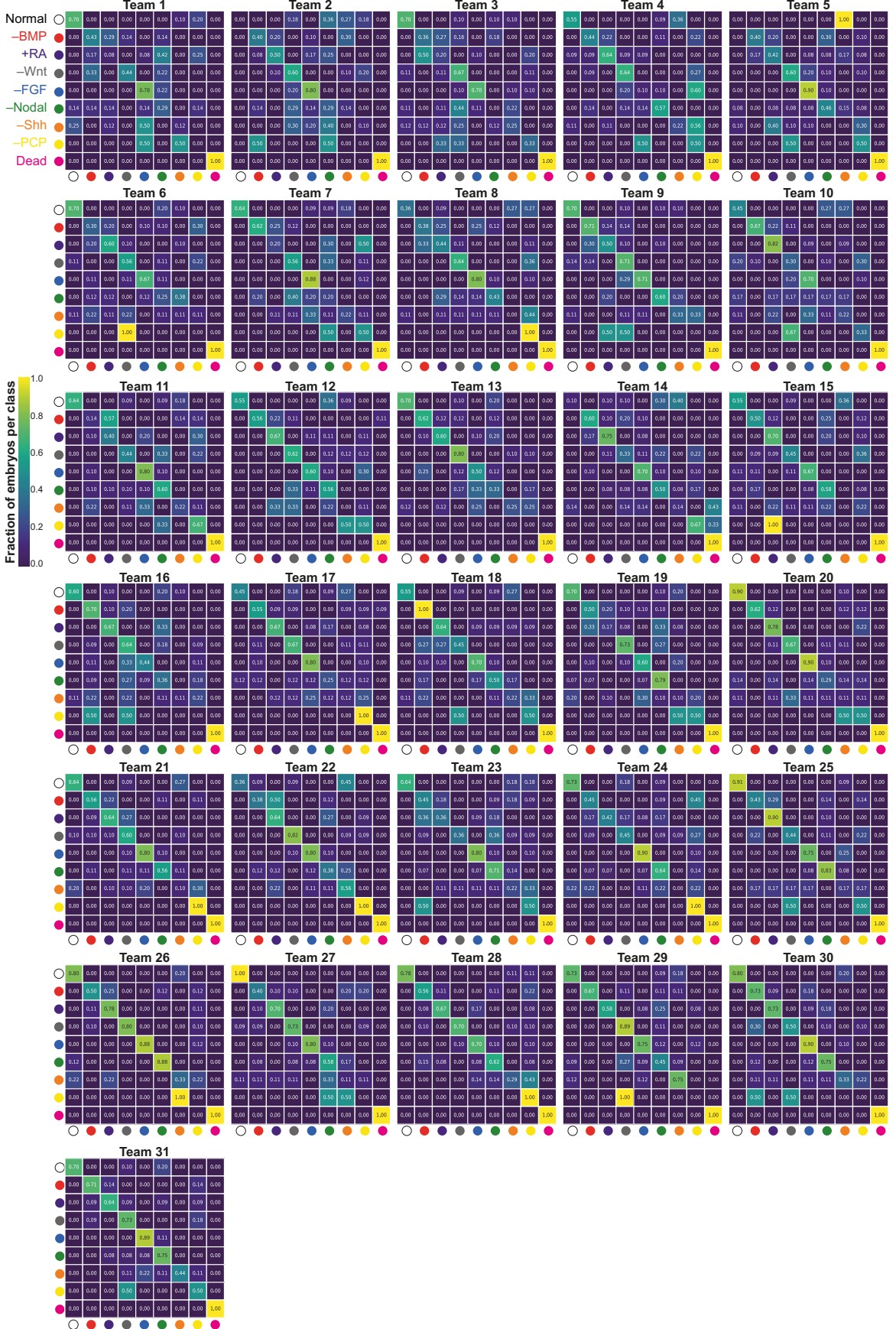

**Extended Data Fig. 2 | Complete data of the non-expert teams to assess 98 embryo images without time information.** Confusion matrices show the classification of the respective non-expert teams. The non-expert teams did not have extra information about the age of the displayed embryos.

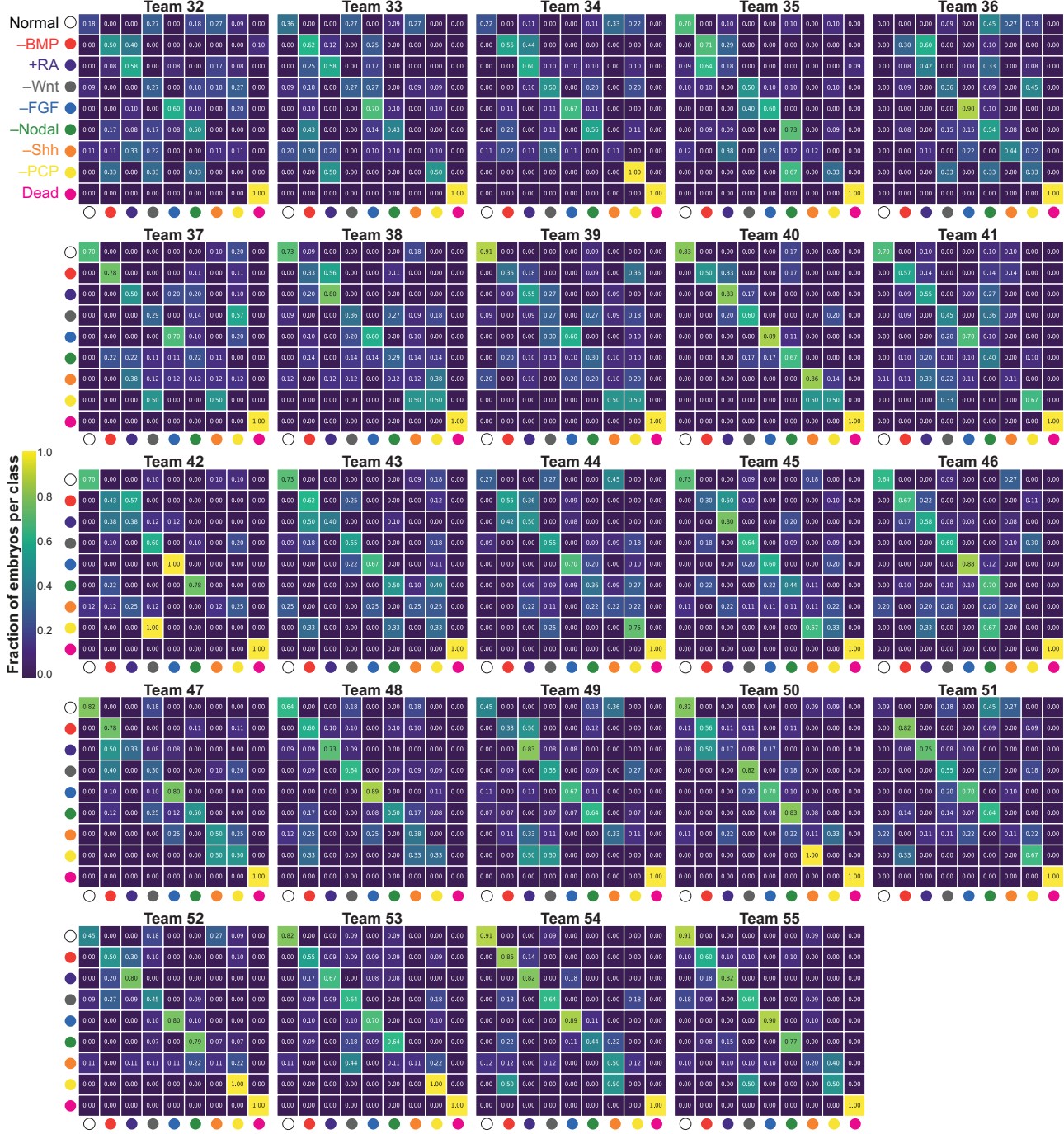

**Extended Data Fig. 3 | Complete data of the non-expert teams to assess 98 embryo images with time information.** Confusion matrices show the classification of the respective teams. Embryo age was supplied with the images.

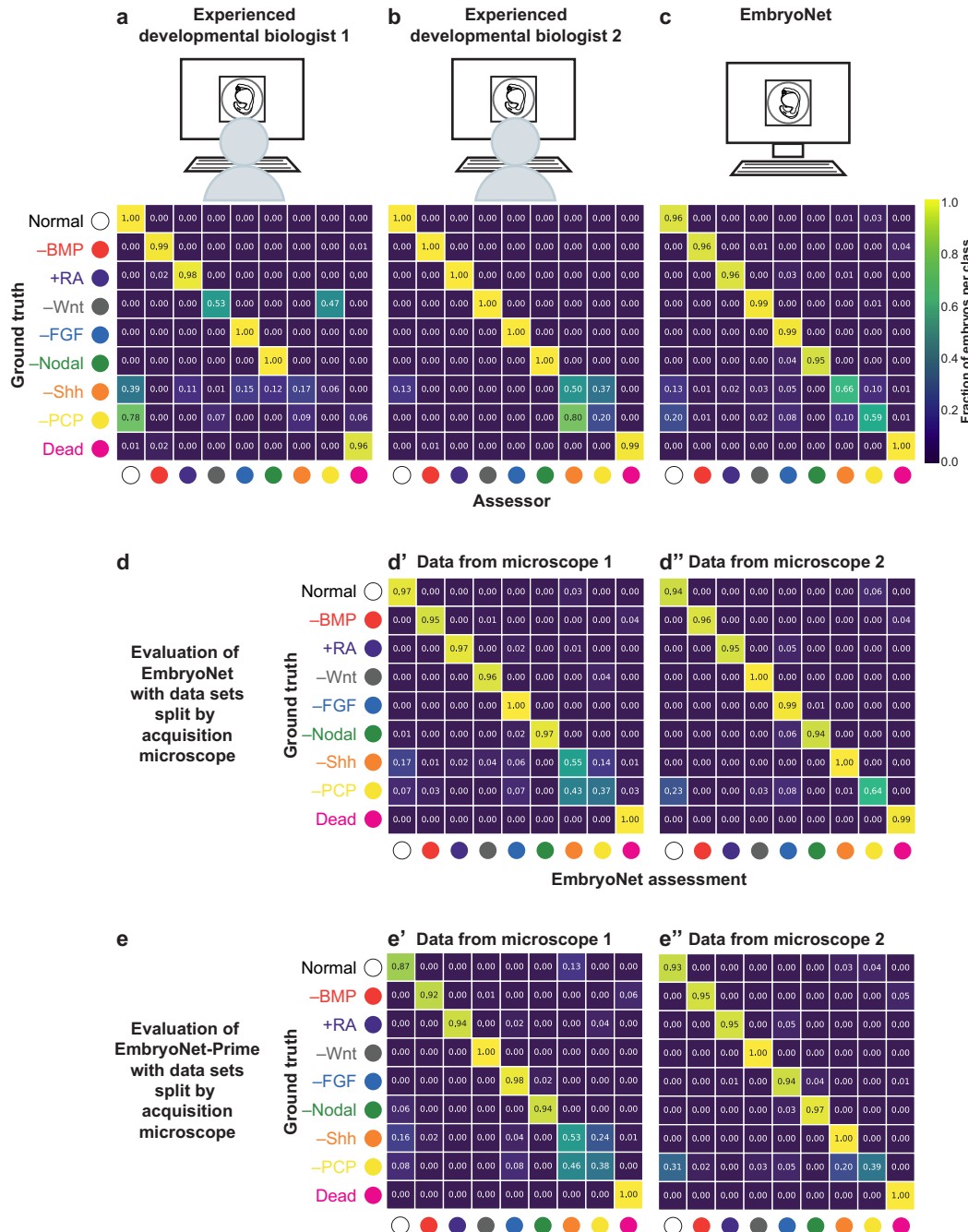

**Extended Data Fig. 4 | Classification of time-lapse data by experts and EmbryoNet, and classification performance with different microscope data sets. (a-c)** Schematics and confusion matrices show classification of image series of the respective assessors. Human experts (a,b) knew that all embryos within a movie received the same treatment. **(d,e)** Confusion matrices showing the performance of EmbryoNet (d) and EmbryoNet-Prime (e) if only test data from either microscope 1 (d',e', ACQUIFER Imaging Machine) or microscope 2 (d'',e'', Keyence BZ-X810) was used.

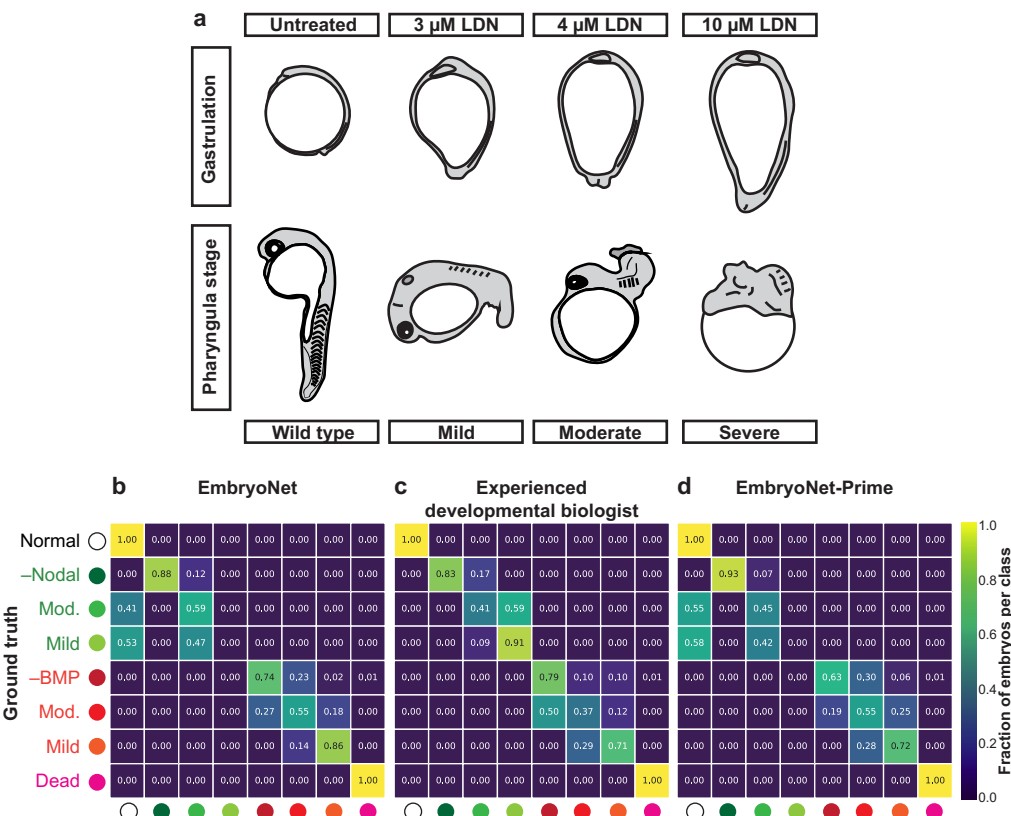

**Extended Data Fig. 5 | Classification of mild Nodal and BMP phenotypes.**
**(a)** Schematic of the experiment with lower inhibitor doses. Lower doses of
the BMP inhibitor LDN-193189 lead to weaker phenotypes, detectable from
late gastrulation onwards. While in the severe cases no clear structures are
distinguishable, moderate embryos have a head and somites and display the
characteristic BMP loss-of-function phenotype with curled-up tails (Kishimoto
et al. 1997). Mild embryos have a largely intact body axis only missing the tail.

**(b-d)** Confusion matrices showing the performance of classification of weaker
phenotypes. EmbryoNet (b) had a lower performance on milder compared to
severe phenotypes, especially for Nodal-inhibited samples. Human annotators
were also less consequent as seen from the confusion matrix between the
accepted ground truth and a second labeler (c). EmbryoNet-Prime had better
success in detecting weak Nodal phenotypes compared to EmbryoNet, but was
less performant on –BMP and on average (d).

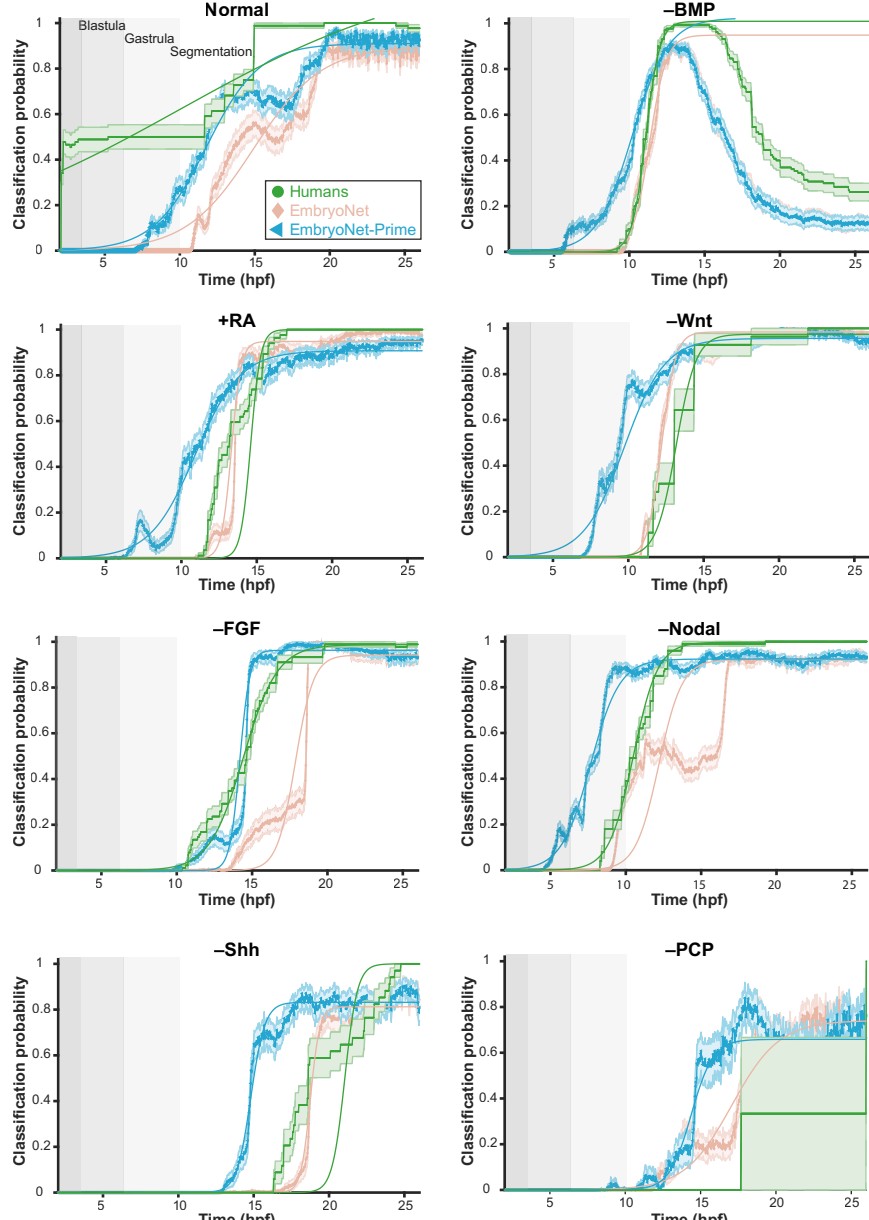

**Extended Data Fig. 6 | Detection time points of all classes by EmbryoNet-Prime.** Plots of the number of detected phenotypes (dotted lines) for each class over time for EmbryoNet (pink diamonds), EmbryoNet-Prime (blue triangles) and human experts (green dots). The error envelopes show standard error of the mean. Solid lines show the fit of a sigmoid curve to the data. Gray boxes show major developmental periods. Different classes could be detected at different time points. −BMP, + RA, −Wnt, −Nodal and −Shh could be classified earlier by EmbryoNet-Prime than by humans. n[Normal]: 74, n[−BMP]: 119, n[+RA]: = 66, n[−Wnt]: 70, n[−FGF]: 74, n[−Nodal]: 110, n[−Shh]: 63, n[−PCP]: 57. Data also shown in Fig. 2g.

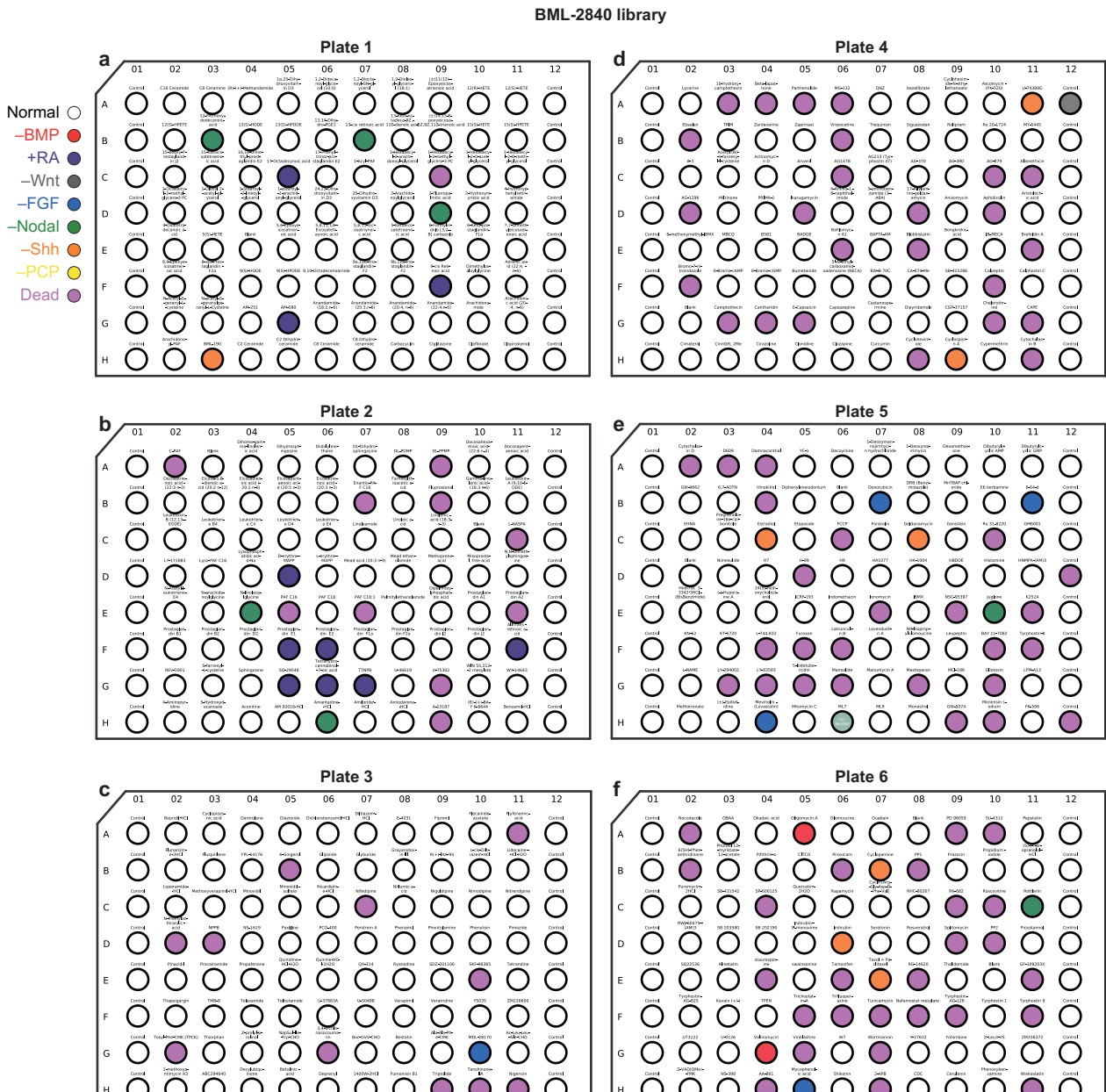

**Extended Data Fig. 7 | Results of the Enzo ScreenWell 2840 library drug screen. (a–f)** Layout of plates 1 (**a**), 2 (**b**), 3 (**c**), 4 (**d**), 5 (**e**) and 6 (**f**) from the BML-2840 library with classifications per well by majority phenotype. See Supplementary Tables 25–30 for details.

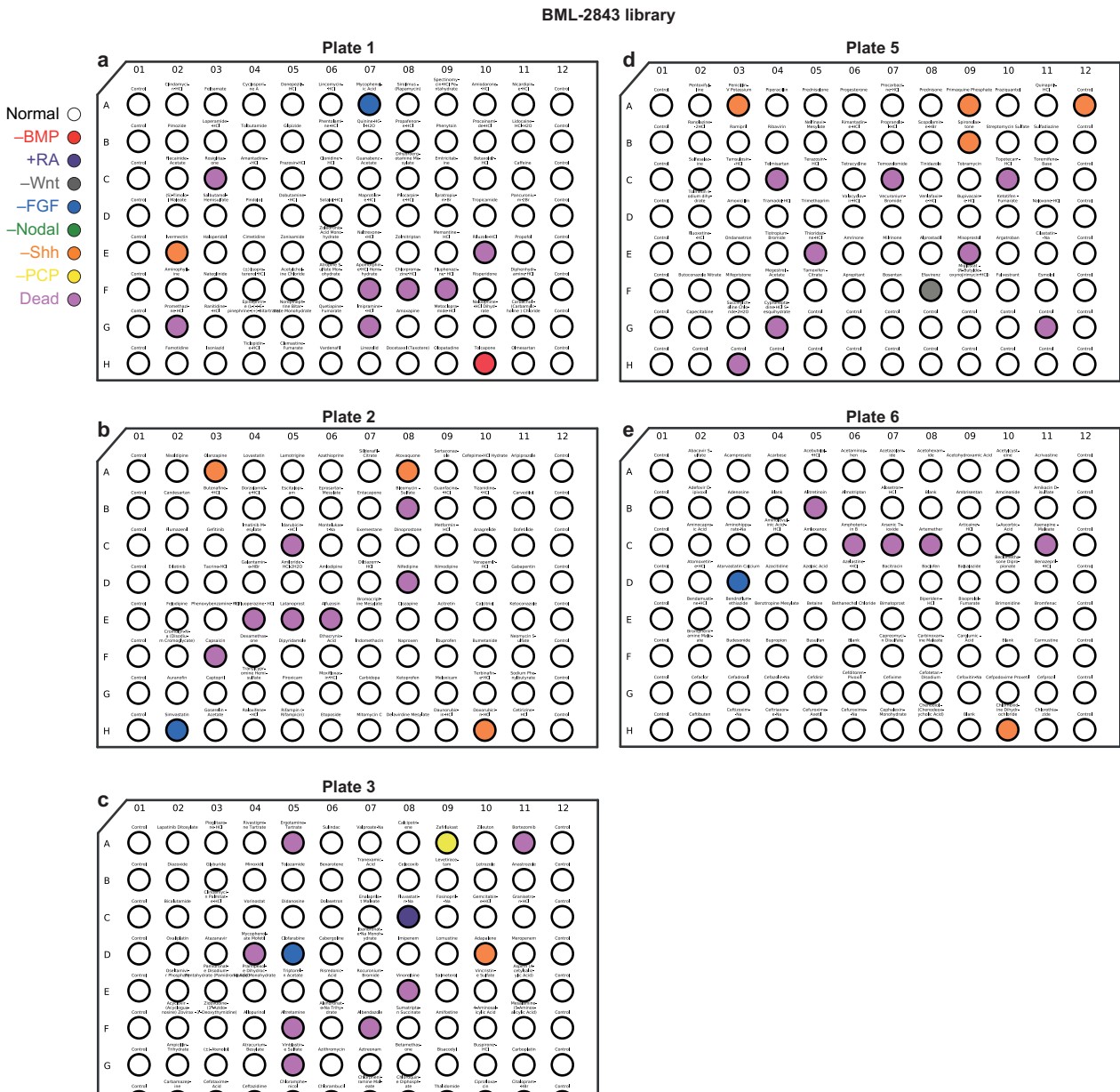

**Extended Data Fig. 8 | Results of the Enzo ScreenWell 2843 library drug screen.** (**a**–**e**) Layout of plates 1 (**a**), 2 (**b**), 3 (**c**), 5 (**d**) and 6 (**e**) from the BML-2843 library with classifications per well by majority phenotype. See Supplementary Tables 31–35 for details.

| | Corresponding author(s): | Patrick Müller |
| --- | --- | --- |
| | Last updated by author(s): | Mar 22, 2023 |

# Reporting Summary

## Statistics

For all statistical analyses, confirm that the following items are present in the figure legend, table legend, main text, or Methods section.

| n/a | Confirmed | |
| --- | --- | --- |
| ☐ | ☒ | The exact sample size (*n*) for each experimental group/condition, given as a discrete number and unit of measurement |
| ☐ | ☒ | A statement on whether measurements were taken from distinct samples or whether the same sample was measured repeatedly |
| ☒ | ☐ | The statistical test(s) used AND whether they are one- or two-sided *Only common tests should be described solely by name; describe more complex techniques in the Methods section.* |
| ☒ | ☐ | A description of all covariates tested |
| ☒ | ☐ | A description of any assumptions or corrections, such as tests of normality and adjustment for multiple comparisons |
| ☐ | ☒ | A full description of the statistical parameters including central tendency (e.g. means) or other basic estimates (e.g. regression coefficient) AND variation (e.g. standard deviation) or associated estimates of uncertainty (e.g. confidence intervals) |
| ☒ | ☐ | For null hypothesis testing, the test statistic (e.g. *F*, *t*, *r*) with confidence intervals, effect sizes, degrees of freedom and *P* value noted *Give P values as exact values whenever suitable.* |
| ☒ | ☐ | For Bayesian analysis, information on the choice of priors and Markov chain Monte Carlo settings |
| ☒ | ☐ | For hierarchical and complex designs, identification of the appropriate level for tests and full reporting of outcomes |
| ☒ | ☐ | Estimates of effect sizes (e.g. Cohen's *d*, Pearson's *r*), indicating how they were calculated |

*Our web collection on statistics for biologists contains articles on many of the points above.*

## Software and code

Policy information about availability of computer code

| Data collection | For data acquisition on an Acquifer Imaging Machine, we used the Imaging Machine control software (Acquifer, Version ID 4.00.21). For data collection on a Keyence BZ-X810 microscope, we used the BZ-X800 viewer (Keyence, Version 01.03.00.01). For data acquisition on a ZEISS Lightsheet Z.1 microscope, we used ZEN 3.1 Black Edition (ZEISS). |
| --- | --- |
| Data analysis | For image annotation, model training, validation and testing we used our custom software EmbryoNet (http://github.com/mueller-lab/EmbryoNet, http://doi.org/10.5281/zenodo.7531593). For the analysis of pErk stainings, we used Fiji/ImageJ version 1.53 and MATLAB (R2022a). The training was performed on an NVIDIA RTX 3090 card in Ubuntu 20.04.4 LTS. |

For manuscripts utilizing custom algorithms or software that are central to the research but not yet described in published literature, software must be made available to editors and reviewers. We strongly encourage code deposition in a community repository (e.g. GitHub). See the Nature Portfolio guidelines for submitting code & software for further information.

# Data

Policy information about availability of data

All manuscripts must include a data availability statement. This statement should provide the following information, where applicable:
- Accession codes, unique identifiers, or web links for publicly available datasets
- A description of any restrictions on data availability
- For clinical datasets or third party data, please ensure that the statement adheres to our policy

Training and evaluation data sets for EmbryoNet are available from http://embryonet.uni-konstanz.de and http://doi.org/10.48606/15. The drug screen data is available from http://doi.org/10.48606/37, http://doi.org/10.48606/38 and http://doi.org/10.48606/41. Additional data that support the findings of this study are available from http://doi.org/10.48606/53 and http://doi.org/10.48606/55.

# Human research participants

Policy information about studies involving human research participants and Sex and Gender in Research.

| Reporting on sex and gender | N/A |
| --- | --- |
| Population characteristics | N/A |
| Recruitment | N/A |
| Ethics oversight | N/A |

Note that full information on the approval of the study protocol must also be provided in the manuscript.

# Field-specific reporting

Please select the one below that is the best fit for your research. If you are not sure, read the appropriate sections before making your selection.

☒ Life sciences          ☐ Behavioural & social sciences          ☐ Ecological, evolutionary & environmental sciences

For a reference copy of the document with all sections, see nature.com/documents/nr-reporting-summary-flat.pdf

# Life sciences study design

All studies must disclose on these points even when the disclosure is negative.

| Sample size | To determine a suitable sample size for the development of EmbryoNet, we used an active learning approach. In an iterative process, we progressively increased the number of images/embryos used as training and validation sets until the classification performance on the validation data set reached a saturation level. This approach is commonly used in classification problems and allows to obtain good classification performance. For all other experiments, at least three biological replicates were estimated to provide an adequate sample size based on previous analyses (Pomreinke et al. eLife 2017, Soh et al. Cell Reports 2020, Kuhn et al. Nature Communications 2022). |
| --- | --- |
| Data exclusions | Embryos that were only partially visible in images were excluded. For the analysis of pERK, only images of embryos that were oriented with the dorsal side facing the camera were used. The dorsal side could be identified after generating maximum intensity projections from image stacks. Embryos with tilted dorso-ventral axes were excluded. |
| Replication | The classification experiment with mixed MZoep and WT embryos comprises four biological replicates performed on the same day with multiple embryos. The heterozygous swirl mutant incross experiment has six biological replicates performed on the same day with multiple embryos. The lft1 and chordin overexpression experiments consist of nine biological replicates performed on the same day with multiple embryos. The classification of a stack of 98 selected embryos was performed once for random classification, 55 times by non-expert teams (31 without time information, 24 with extra time information), and by one experienced developmental biologist. The classification of time-lapse data sets was performed by two experienced developmental biologists once each. All automatic classifications are from one repetition by EmbryoNet or by EmbryoNet-Prime, respectively. The drug screen was performed once with multiple embryos for each treatment. The Statin findings were confirmed twice independently on separate days. The pERK immunostainings were performed with multiple embryos on the same day. |
| Randomization | Embryos from zebrafish and medaka crosses as well as stickleback in vitro-fertilizations were randomly allocated into experimental groups. |
| Blinding | Since embryos from zebrafish and medaka crosses as well as stickleback in-vitro fertilizations were genetically uniform and indistinguishable, blinding of the investigators was not necessary. |

# Reporting for specific materials, systems and methods

We require information from authors about some types of materials, experimental systems and methods used in many studies. Here, indicate whether each material, system or method listed is relevant to your study. If you are not sure if a list item applies to your research, read the appropriate section before selecting a response.

## Materials & experimental systems

| n/a | Involved in the study |
|-----|----------------------|
| ☐ | ☒ Antibodies |
| ☒ | ☐ Eukaryotic cell lines |
| ☒ | ☐ Palaeontology and archaeology |
| ☐ | ☒ Animals and other organisms |
| ☒ | ☐ Clinical data |
| ☒ | ☐ Dual use research of concern |

## Methods

| n/a | Involved in the study |
|-----|----------------------|
| ☒ | ☐ ChIP-seq |
| ☒ | ☐ Flow cytometry |
| ☒ | ☐ MRI-based neuroimaging |

## Antibodies

| | |
|---|---|
| Antibodies used | We used anti-DP-ERK (Sigma-Aldrich, M8159) antibody at a dilution of 1:5000, and HRP-conjugated anti-mouse (Jackson ImmunoResearch, 715-035-150) antibody at a dilution of 1:5000. |
| Validation | We used a validated primary antibody from a standard commercial source (for validation see for example Rogers et al. eLife 2020, Navon et al. J Mol Neurosci 2012, Larbuisson et al. Differentiation 2013). The secondary antibody was also from a standard commercial source (for validation see e.g. Kim et al. Nature Communications 2022, Rogers et al. eLife 2020, Pardi et al. Nature Communications 2022). |

## Animals and other research organisms

Policy information about studies involving animals; ARRIVE guidelines recommended for reporting animal research, and Sex and Gender in Research

| | |
|---|---|
| Laboratory animals | The experiments were performed exclusively with embryos and larvae that were not yet freely feeding. We used wild-type zebrafish, medaka and stickleback embryos. In addition, swirl mutants (Kishimoto et al. 1997) and maternal-zygotic oep zebrafish mutants were used (Gritsman et al. Cell 1999). Age of embryos: zebrafish (TE, oep and swr strains): 0-48 hpf; medaka (Cab strain): 0-48 hpf; stickleback (Little Campbell River and Tyne River strains): 0-140 hpf. |
| Wild animals | We did not use wild animals. |
| Reporting on sex | Sex-based analysis was not performed because phenotypical sex identification is not possible in zebrafish, medaka or stickleback embryos. |
| Field-collected samples | We did not use field-collected samples. |
| Ethics oversight | All procedures were executed in accordance with the guidelines of the EU directive 2010/63/EU and the German Animal Welfare Act as approved by the local authorities represented by the Regierungspräsidium Tübingen and the Regierungspräsidium Freiburg (Baden-Württemberg, Germany). |

Note that full information on the approval of the study protocol must also be provided in the manuscript.

