## [Peer Review File · Nature Methods]

Peer Review Information

Manuscript Title: EmbryoNet: Using deep learning to link embryonic phenotypes to signaling pathways

Corresponding author name(s): Patrick Müller

Editorial Notes:

Reviewer Comments & Decisions:

Decision Letter, initial version:
--

Dear Professor Müller,

Thank you for your letter detailing how you would respond to the reviewer concerns regarding your Resource, "EmbryoNet: Using deep learning to link embryonic phenotypes to signaling pathways". We have decided to invite you to revise your manuscript as you have outlined, before we reach a final decision on publication.

With regards to additional controls for pathway specificity, we think the data you have already should be sufficient for the revision, and we do not ask you to analyze additional strains or perturbations beyond this.

With regards to the dose response query, we think your response in the rebuttal is sufficient, and do not need to see changes to the main text unless this is something you would like to include in the discussion.

- * include a point-by-point response to the reviewers and to any editorial suggestions
- * please underline/highlight any additions to the text or areas with other significant changes to facilitate review of the revised manuscript
- * address the points listed described below to conform to our open science requirements
- * ensure it complies with our general format requirements as set out in our guide to authors at www.nature.com/naturemethods
- * resubmit all the necessary files electronically by using the link below to access your home page [Redacted] This URL links to your confidential home page and associated information about manuscripts you may have submitted, or that you are reviewing for us. If you wish to forward this email to co-authors, please delete the link to your homepage.

We hope to receive your revised paper within 2-3 months. If you cannot send it within this time, please let us know. In this event, we will still be happy to reconsider your paper at a later date so long as nothing similar has been accepted for publication at Nature Methods or published elsewhere.

OPEN SCIENCE REQUIREMENTS

REPORTING SUMMARY AND EDITORIAL POLICY CHECKLISTS

Please note that these forms are dynamic ‘smart pdfs’ and must therefore be downloaded and completed in Adobe Reader. We will then flatten them for ease of use by the reviewers. If you would like to reference the guidance text as you complete the template, please access these flattened versions at <http://www.nature.com/authors/policies/availability.html>.

IMAGE INTEGRITY

DATA AVAILABILITY

Please include a “Data availability” subsection in the Online Methods. This section should inform readers about the availability of the data used to support the conclusions of your study, including accession codes to public repositories, references to source data that may be published alongside the paper, unique identifiers such as URLs to data repository entries, or data set DOIs, and any other statement about data availability. At a minimum, you should include the following statement: “The data that support the findings of this study are available from the corresponding author upon request”, describing which data is available upon request and mentioning any restrictions on availability. If DOIs are provided, please include these in the Reference list (authors, title, publisher (repository name), identifier, year). For more guidance on how to write this section please see: <http://www.nature.com/authors/policies/data/data-availability-statements-data-citations.pdf>

CODE AVAILABILITY

Please include a “Code Availability” subsection in the Online Methods which details how your custom code is made available. Only in rare cases (where code is not central to the main conclusions of the paper) is the statement “available upon request” allowed (and reasons should be specified).

For more information on our code sharing policy and requirements, please see:
<https://www.nature.com/nature-research/editorial-policies/reporting-standards#availability-of-computer-code>

MATERIALS AVAILABILITY

ORCID

Sincerely,

Rita

Rita Strack, Ph.D.
Senior Editor
Nature Methods

Reviewers' Comments:

Reviewer #1:

Remarks to the Author:

This work describes very convincingly how AI can be used to classify developmental defects in zebrafish embryos according to the main signaling pathways that could be affected. Using mutants and specific agonists or antagonists, the authors capture video from the corresponding embryos (in large numbers, and in different positions), to then train their system to classify each embryo according to the given annotations and time stamp of the learning set. The accuracy they reach is at least comparable to that of the experienced biologists, and sometimes better. They then go even further to show that their EmbryoNet is able to perform classification at even earlier time points during development, thereby adding new information to the known phenotypes.

The approach is novel, the results are impressive. EmbryoNet is a valuable addition to the tools that are available for phenotyping mutants, analyzing embryotoxic compounds, and increasing speed and throughput for this type of experiments.

The methods and the results are clearly presented, reading is straightforward. The conclusions are clearly stated and robust.

I have very little comments to make:

- Line 357 : " To generalize the method independently of the microscope,": Were the two microscopes used for each learning set, all the time? Or were the images from each microscope treated independently? In other words, did each learning set always include videos from both microscopes, and in what proportion? Did each microscope cover the entire range of phenotypes in the set?

- Lines 272-274: "We were also able to adapt EmbryoNet for assessing other fish species separated from zebrafish by hundreds of millions of years in evolution, enabling the analysis of high-dimensional phenomic data across taxa." This statement needs somewhat of a clarification. I understand that a new training was needed for analyzing medaka and stickleback, with obviously good results, however it is difficult to see how this would help "the analysis of high-dimensional phenomic data across taxa". In my view, that would only be the case if you could transfer the trained models directly through adjusting the developmental timing.

- Line 278: "compounds" instead of "compunds"

Reviewer #2:

Remarks to the Author:

The Manuscript by Čapek describes an elegant computational approach to detect signaling specific phenotypes in embryonic fish. The authors validate this tool and employ it in a drug screen to link drug induced phenotypes to FGF signalling, a novel finding. The model is also used to detect specific phenotypes much earlier than humanly possible and performs well, when re-trained and transferred to other fish species.

I enjoyed reading the manuscript and found the approach to be innovative, systematic and well executed. Together with the very open way of sharing the experimental data, good descriptions of the procedures performed and additional online resources, such a GUI interface, I find this to be a very fitting and adequate paper for Nature Methods. I also find the unconventional training process to be well described and informative.

A few comments:

The paper is focused on detecting interference with the main signalling pathways active during early embryogenesis. Many pharmacological treatments or genetic manipulation, however, will result in phenotypes that are distinct from these signalling disruptions or even interfere with multiple of these. To what degree would EmbryoNet handle such situations? Would this result in misclassification into one of the signalling categories? As I don't see a "abnormâl otherwise" in the drug screening experiment, for example. I appreciate that it is impossible to train for such ill defined classes, but a better understanding of this issue would be of great value for judging how applicable this model will be.

A number of signalling interferences were done pharmacologically or by injection of Morpholinos or mRNA. The authors nicely evaluate the nodal phenotype with genetic knockout like, but the other pathways are less well confirmed to be specific. Also, morpholinos are known to elicit unspecific or off-target effects, also here, additional controls would be desirable.

The observation that EmbryoNet-Prime picks up phenotypes correctly much earlier than human observers is striking and very compelling. Is this similar for dose responses of the drug treatments? Testing, if the threshold for a drug dose is lower for the network than human experts would be interesting.

The finding that statins interfere with FGF signalling is interesting, but to make such a strong conclusion, additional functional experiments beyond the pERK staining are required.

The cross-species employment of the network is a strong additional finding of the paper. However, it seems the original EmbyroNet was re-trained to accommodate the species specific differences. However, in Fig. 4 f, g, the label EmbryoNet implies that the existing network was employed without modification. I suggest to clarify this point to avoid any wrong impressions.

Was indeed an RTX 3900 used or a RTX 3090?

Author Rebuttal to Initial comments

Response to reviewers' comments

We thank the reviewers for their helpful and constructive criticism. We have highlighted the changes in the revised manuscript in blue font. Below, we address each point raised by the reviewers individually.

Reviewer #1:

Remarks to the Author:

This work describes very convincingly how AI can be used to classify developmental defects in zebrafish embryos according to the main signaling pathways that could be affected. Using mutants and specific agonists or antagonists, the authors capture video from the corresponding embryos (in large numbers, and in different positions), to then train their system to classify each embryo according to the given annotations and time stamp of the learning set. The accuracy they reach is at least comparable to that of the experienced biologists, and sometimes better. They then go even further to show that their EmbryoNet is able to perform classification at even earlier time points during development, thereby adding new information to the known phenotypes.

The approach is novel, the results are impressive. EmbryoNet is a valuable addition to the tools that are available for phenotyping mutants, analyzing embryotoxic compounds, and increasing speed and throughput for this type of experiments.

The methods and the results are clearly presented, reading is straightforward. The conclusions are clearly stated and robust.

I have very little comments to make:

- Line 357 : “ To generalize the method independently of the microscope, ...”: Were the two microscopes used for each learning set, all the time? Or were the images from each microscope treated independently? In other words, did each learning set always include videos from both microscopes, and in what proportion? Did each microscope cover the entire range of phenotypes in the set?

The two microscopes were used for each learning set, all the time. Each learning set always included videos from both microscopes: 76% of the images from one microscope (ACQUIFER Imaging Machine) and 24% from the other (Keyence BZ-X810). Each microscope covered the entire range of phenotypes in the set. We included the data for these statistics in Supplementary Tables 2, 14-15 and 23-24 of the revised manuscript.

To evaluate the performance of EmbryoNet, we have also generated independent validation data sets for each microscope. We found that the performance of EmbryoNet was similar on these data sets as shown in the confusion matrices below. We have included this comparison in Supplementary Fig. 4 of the revised manuscript.

- Lines 272-274: “We were also able to adapt EmbryoNet for assessing other fish species separated from zebrafish by hundreds of millions of years in evolution, enabling the analysis of high-dimensional phenomic data across taxa.” This statement needs somewhat of a clarification. I understand that a new training was needed for analyzing medaka and stickleback, with obviously good results, however it is difficult to see how this would help “the analysis of high-dimensional phenomic data across taxa”. In my view, that would only be the case if you could transfer the trained models directly through adjusting the developmental timing.

This point was also raised by Reviewer #2, and we thank both reviewers for pointing out the need for clarification. In the revised manuscript, we have clarified this point as follows: “*We were also able to retrain EmbryoNet for assessing other fish species separated from zebrafish by hundreds of millions of years in evolution, enabling the analysis of high-dimensional phenomic data in different taxa*”.

- Line 278: “compounds” instead of “compunds”

Thank you for pointing out the typo – we have corrected this oversight in the revised manuscript.

Reviewer #2:

Remarks to the Author:

The Manuscript by Čapek describes an elegant computational approach to detect signaling specific phenotypes in embryonic fish. The authors validate this tool and employ it in a drug screen to link drug induced phenotypes to FGF signalling, a novel finding. The model is also used to detect specific phenotypes much earlier than humanly possible and performs well, when re-trained and transferred to other fish species.

I enjoyed reading the manuscript and found the approach to be innovative, systematic and well executed. Together with the very open way of sharing the experimental data, good descriptions of the procedures performed and additional online resources, such a GUI interface, I find this to be a very fitting and adequate paper for Nature Methods. I also find the unconventional training process to be well described and informative.

A few comments:

The paper is focused on detecting interference with the main signalling pathways active during early embryogenesis. Many pharmacological treatments or genetic manipulation, however, will result in phenotypes that are distinct from these signalling disruptions or even interfere with multiple of these. To what degree would EmbryoNet handle such situations? Would this result in misclassification into one of the signalling categories? As I don't see a “abnormal otherwise” in the drug screening experiment, for example. I appreciate that it is impossible to train for such ill defined classes, but a better understanding of this issue would be of great value for judging how applicable this model will be.

EmbryoNet reports a probability for each assessment of a phenotype. These probabilities are visualized in Supplementary Movies 9-24 and reach nearly 100% for a single class if an embryo was left untreated or if a signaling pathway was modulated with a pathway-specific treatment. In case of other disruptions or mixed phenotypes – e.g. resulting from the alteration of more than one signaling pathway – we expect that these probabilities will be distributed across several phenotype classes. This is likely to capture phenotypic features that have their root in

signaling modulations, and EmbryoNet can provide hypotheses about which signaling pathways to analyze in further re-tests.

Misclassification can result from images, in which embryos were transiently in unfavorable positions. To account for these misclassifications, we used several embryos per well in our drug screen, calculated the total number of images in which a phenotype was detected by EmbryoNet, and then calculated the fraction of the most frequent phenotype class as a measure of confidence that a certain signaling pathway was perturbed. For known signaling pathway modulators in our drug library, this measure of confidence was frequently near 100%, e.g. 9-cis retinoic acid: 100% (Supplementary Table 25), All-trans retinoic acid: 100% (Supplementary Table 26), Cyclopamine: 99% (Supplementary Table 30). For other small-molecule treatments in our library, this confidence measure can be lower if EmbryoNet classifies the images into different categories, providing hypotheses about which signaling pathways to analyze in further re-tests.

A number of signalling interferences were done pharmacologically or by injection of Morpholinos or mRNA. The authors nicely evaluate the nodal phenotype with genetic knockout likes, but the other pathways are less well confirmed to be specific. Also, morpholinos are known to elicit unspecific or off-target effects, also here, additional controls would be desirable.

We agree that it is important to ensure that the treatments for our class definitions are specific. We would like to point out that the small molecules to modulate RA (e.g. D'Aniello et al., 2015; Dalgin et al., 2011; Franzosa et al., 2013), FGF (e.g. Lovely et al., 2016; Maier and Whitfield, 2014; Osborn et al., 2020; Rohner et al., 2009; Sun et al., 1999), Wnt (e.g. Grainger et al., 2016; Kamei et al., 2019; Nie et al., 2019; Takayama et al., 2018; Wang et al., 2013), Nodal (e.g. DaCosta Byfield et al., 2004; Deshwar et al., 2016; Gonsar et al., 2016; Terashima et al., 2014; van Boxel et al., 2018), BMP (e.g. Cannon et al., 2010; Cuny et al., 2008; Kruse-Bend et al., 2012; Place and Smith, 2017; Steinbicker et al., 2011; Zinck et al., 2021) and Shh signaling (e.g. Incardona et al., 1998; Muthu et al., 2016; Osborn et al., 2020; Quint et al., 2002) have been validated for specificity and widely applied in previous studies. mRNA injections of pathway antagonists also induce *bona fide* signaling pathway loss-of-function phenotypes, and ectopically provided *Lefty* and *Chordin* mRNA can even rescue the respective zebrafish mutants (e.g. Fisher and Halpern, 1999; Rogers et al., 2017; Schulte-Merker et al., 1997; Tuazon et al., 2020). We would also like to point out that, while some morpholinos can elicit unspecific or off-target effects, the morpholino to induce the -PCP phenotype in our manuscript has been extensively validated in previous studies by leading labs in the PCP field (e.g. Johansson et al., 2019; Love et al., 2018; Prince and Jessen, 2019; Williams et al., 2012; Williams and Solnica-Krezel, 2020).

To further validate our approach, we directly compared phenotypes induced by small-molecule inhibitors, pathway antagonists or mutants.

These were then classified by EmbryoNet. The figure to the left shows Nodal phenotypes induced by small-molecule inhibitor treatment (SB-505124, n=33), injection of a pathway antagonist (*Lefty* mRNA, n=27), or in a receptor mutant (MZoep, n=27), which were all classified as -Nodal with similar accuracy.

We have extended this approach for BMP signaling as well. The figure to the right shows BMP phenotypes induced by small-molecule inhibitor treatment (LDN-193189, $n=45$), pathway antagonist injection (*Chordin* mRNA, $n=26$), or in a pathway ligand mutant (*swirl*^{-/-}, $n=13$), which were all classified by EmbryoNet as -BMP with similar accuracy. Importantly, -BMP phenotypes generated by overexpression of the BMP inhibitor Chordin were properly identified by EmbryoNet, even though such treatments had not been used for the training of the network. Furthermore, EmbryoNet recognized -*Shh* phenotypes with similar accuracy for both small-molecule treatment using cyclopamine (82%) and *zGli3R-GFP* mRNA injection (72%). We have included these additional analyses in Supplementary Fig. 1 and Supplementary Note 2 of the revised manuscript.

The observation that EmbryoNet-Prime picks up phenotypes correctly much earlier than human observers is striking and very compelling. Is this similar for dose responses of the drug treatments? Testing, if the threshold for a drug dose is lower for the network than human experts would be interesting.

This is a very interesting possibility, but the experiment is conceptually complicated. When we trained EmbryoNet-Prime, we used information from later stages of developing zebrafish to label the training and test data before the phenotypes could be recognized by a human assessor. If we treat embryos with inhibitor levels that are low enough such that they are not distinguishable from wild-type embryos by eye, human annotators are not confidently able to decide whether the inhibitor levels were too low to have an effect or whether the effect is just invisible to the human eye. These experiments are further complicated by a non-uniform distribution of phenotypes, and it is currently unclear why exactly phenotypes differ in genetically similar embryos that were uniformly bathed in small-molecule inhibitors. Therefore, it is not easily possible to label a reliable ground-truth data set.

To illustrate this point, we treated zebrafish with very low levels of the Nodal inhibitor SB-505124 (4 μ M) and used these data to test if hidden phenotypes might also be detectable. Of a total of 62 embryos, a human expert annotator with knowledge about the treatment classified 43 embryos as mild Nodal phenotypes and 19 as wild-type. For the same data set, EmbryoNet detected 18 embryos as -*Nodal* and 44 as wild-type. While these preliminary experiments are conceptually and technically complicated as pointed out above – and the annotators had prior knowledge about the treatment – this might suggest that the threshold for a drug dose is not lower for EmbryoNet-Prime than human experts.

In the future, we will follow up on the reviewer's interesting suggestion using fine-grained inhibitor titration series combined with unsupervised classification as an alternative approach.

The finding that statins interfere with FGF signalling is interesting, but to make such a strong conclusion, additional functional experiments beyond the pERK staining are required.

Thank you. We would like to point out that dorsal-ventral patterning phenotypes induced by Statins have previously been reported in zebrafish embryos (Campos et al., 2016; Campos et al., 2015, also cited in our manuscript). In retrospect, these defects look like classical -*FGF* phenotypes, but to the best of our knowledge a connection between Statin-induced dorsal-ventral patterning phenotypes and the FGF signaling pathway has not yet been made in the literature. Furthermore, Statins have previously been shown to lower pErk levels in human

endometrial stromal cells (Piotrowski et al., 2006, also cited in our manuscript), and a potential effect of Statins on FGF signaling, as observed in our zebrafish experiments, has previously been discussed for cultured human cells (Burgazli et al., 2014; Fafilek et al., 2017; Park et al., 2006; Shiota et al., 2012; Yamashita et al., 2014).

Our screen independently recovered the Statin-induced phenotype and linked it to FGF signaling. Interestingly, this phenotype and signaling pathway association was independently recovered three times for different statins (Lovastatin, Simvastatin, Atorvastatin), and we were able to show FGF signaling modulation at concentrations similar to those used in current medications (Supplementary Note 4 of the revised manuscript). However, we agree that the bioavailability in zebrafish embryos compared to human cells is currently unclear, and we have added this additional qualifying statement about the limitations of our experiments in the revised manuscript. Finally, we would like to point out that the focus of the current work is to provide a general deep-learning framework to link embryonic phenotypes to signaling pathways, and we will follow up on the molecular link between Statins and FGF signaling as well as further biomedical implications in future work.

The cross-species employment of the network is a strong additional finding of the paper. However, it seems the original EmbryoNet was re-trained to accommodate the species specific differences. However, in Fig. 4 f, g, the label EmbryoNet implies that the existing network was employed without modification. I suggest to clarify this point to avoid any wrong impressions.

This point was also raised by Reviewer #1, and we thank both reviewers for pointing out the need for clarification. In the revised manuscript, we have clarified this point as follows: “*We were also able to retrain EmbryoNet for assessing other fish species separated from zebrafish by hundreds of millions of years in evolution, enabling the analysis of high-dimensional phenomic data in different taxa*”.

Was indeed an RTX 3900 used or a RTX 3090?

It was indeed an RTX 3090 – we have corrected this oversight in the revised manuscript.

References

- Burgazli, K.M., Behrendt, M.A., Mericililer, M., Chasan, R., Parahuleva, M., and Erdogan, A. (2014). The impact of statins on FGF-2-stimulated human umbilical vein endothelial cells. *Postgrad Med J* 126, 118-128.
- Campos, L.M., Rios, E.A., Guapyassu, L., Midlej, V., Atella, G.C., Herculano-Houzel, S., Benchimol, M., Mermelstein, C., and Costa, M.L. (2016). Alterations in zebrafish development induced by simvastatin: Comprehensive morphological and physiological study, focusing on muscle. *Exp Biol Med (Maywood)* 241, 1950-1960.
- Campos, L.M., Rios, E.A., Midlej, V., Atella, G.C., Herculano-Houzel, S., Benchimol, M., Mermelstein, C., and Costa, M.L. (2015). Structural analysis of alterations in zebrafish muscle differentiation induced by simvastatin and their recovery with cholesterol. *J Histochem Cytochem* 63, 427-437.
- Cannon, J.E., Upton, P.D., Smith, J.C., and Morrell, N.W. (2010). Intersegmental vessel formation in zebrafish: requirement for VEGF but not BMP signalling revealed by selective and non-selective BMP antagonists. *Br J Pharmacol* 161, 140-149.

Cuny, G.D., Yu, P.B., Laha, J.K., Xing, X., Liu, J.F., Lai, C.S., Deng, D.Y., Sachidanandan, C., Bloch, K.D., and Peterson, R.T. (2008). Structure-activity relationship study of bone morphogenetic protein (BMP) signaling inhibitors. *Bioorg Med Chem Lett* *18*, 4388-4392.

D'Aniello, E., Ravisankar, P., and Waxman, J.S. (2015). Rdh10a Provides a Conserved Critical Step in the Synthesis of Retinoic Acid during Zebrafish Embryogenesis. *PLoS One* *10*, e0138588.

DaCosta Byfield, S., Major, C., Laping, N.J., and Roberts, A.B. (2004). SB-505124 is a selective inhibitor of transforming growth factor-beta type I receptors ALK4, ALK5, and ALK7. *Mol Pharmacol* *65*, 744-752.

Dalgin, G., Ward, A.B., Hao le, T., Beattie, C.E., Nechiporuk, A., and Prince, V.E. (2011). Zebrafish *mnx1* controls cell fate choice in the developing endocrine pancreas. *Development* *138*, 4597-4608.

Deshwar, A.R., Chng, S.C., Ho, L., Reversade, B., and Scott, I.C. (2016). The Apelin receptor enhances Nodal/TGFbeta signaling to ensure proper cardiac development. *Elife* *5*.

Fafilek, B., Hampl, M., Ricankova, N., Vesela, I., Balek, L., Kunova Bosakova, M., Gudemova, I., Varecha, M., Buchtova, M., and Krejci, P. (2017). Statins do not inhibit the FGFR signaling in chondrocytes. *Osteoarthritis Cartilage* *25*, 1522-1530.

Fisher, S., and Halpern, M.E. (1999). Patterning the zebrafish axial skeleton requires early chordin function. *Nat Genet* *23*, 442-446.

Franzosa, J.A., Bugel, S.M., Tal, T.L., La Du, J.K., Tilton, S.C., Waters, K.M., and Tanguay, R.L. (2013). Retinoic acid-dependent regulation of miR-19 expression elicits vertebrate axis defects. *FASEB J* *27*, 4866-4876.

Gonsar, N., Coughlin, A., Clay-Wright, J.A., Borg, B.R., Kindt, L.M., and Liang, J.O. (2016). Temporal and spatial requirements for Nodal-induced anterior mesendoderm and mesoderm in anterior neurulation. *Genesis* *54*, 3-18.

Grainger, S., Richter, J., Palazon, R.E., Pouget, C., Lonquich, B., Wirth, S., Grassme, K.S., Herzog, W., Swift, M.R., Weinstein, B.M., *et al.* (2016). Wnt9a Is Required for the Aortic Amplification of Nascent Hematopoietic Stem Cells. *Cell Rep* *17*, 1595-1606.

Incardona, J.P., Gaffield, W., Kapur, R.P., and Roelink, H. (1998). The teratogenic Veratrum alkaloid cyclopamine inhibits sonic hedgehog signal transduction. *Development* *125*, 3553-3562.

Johansson, M., Giger, F.A., Fielding, T., and Houart, C. (2019). Dkk1 Controls Cell-Cell Interaction through Regulation of Non-nuclear beta-Catenin Pools. *Dev Cell* *51*, 775-786 e773.

Kamei, C.N., Gallegos, T.F., Liu, Y., Hukriede, N., and Drummond, I.A. (2019). Wnt signaling mediates new nephron formation during zebrafish kidney regeneration. *Development* *146*.

Kruse-Bend, R., Rosenthal, J., Quist, T.S., Veien, E.S., Fuhrmann, S., Dorsky, R.I., and Chien, C.B. (2012). Extraocular ectoderm triggers dorsal retinal fate during optic vesicle evagination in zebrafish. *Dev Biol* *371*, 57-65.

Love, A.M., Prince, D.J., and Jessen, J.R. (2018). Vangl2-dependent regulation of membrane protrusions and directed migration requires a fibronectin extracellular matrix. *Development* *145*.

Lovely, C.B., Swartz, M.E., McCarthy, N., Norrie, J.L., and Eberhart, J.K. (2016). Bmp signaling mediates endoderm pouch morphogenesis by regulating Fgf signaling in zebrafish. *Development* *143*, 2000-2011.

- Maier, E.C., and Whitfield, T.T. (2014). RA and FGF signalling are required in the zebrafish otic vesicle to pattern and maintain ventral otic identities. *PLoS Genet* *10*, e1004858.
- Muthu, V., Eachus, H., Ellis, P., Brown, S., and Placzek, M. (2016). Rx3 and Shh direct anisotropic growth and specification in the zebrafish tuberal/anterior hypothalamus. *Development* *143*, 2651-2663.
- Nie, C.H., Wan, S.M., Liu, Y.L., Liu, H., Wang, W.M., and Gao, Z.X. (2019). Development of Teleost Intermuscular Bones Undergoing Intramembranous Ossification Based on Histological-Transcriptomic-Proteomic Data. *Int J Mol Sci* *20*.
- Osborn, D.P.S., Li, K., Cutty, S.J., Nelson, A.C., Wardle, F.C., Himits, Y., and Hughes, S.M. (2020). Fgf-driven Tbx protein activities directly induce myf5 and myod to initiate zebrafish myogenesis. *Development* *147*.
- Park, H.J., Zhang, Y., Georgescu, S.P., Johnson, K.L., Kong, D., and Galper, J.B. (2006). Human umbilical vein endothelial cells and human dermal microvascular endothelial cells offer new insights into the relationship between lipid metabolism and angiogenesis. *Stem Cell Rev* *2*, 93-102.
- Piotrowski, P.C., Kwintkiewicz, J., Rzepczynska, I.J., Seval, Y., Cakmak, H., Arici, A., and Duleba, A.J. (2006). Statins inhibit growth of human endometrial stromal cells independently of cholesterol availability. *Biol Reprod* *75*, 107-111.
- Place, E.S., and Smith, J.C. (2017). Zebrafish *atoh8* mutants do not recapitulate morpholino phenotypes. *PLoS One* *12*, e0171143.
- Prince, D.J., and Jessen, J.R. (2019). Dorsal convergence of gastrula cells requires Vangl2 and an adhesion protein-dependent change in protrusive activity. *Development* *146*.
- Quint, E., Smith, A., Avaron, F., Laforest, L., Miles, J., Gaffield, W., and Akimenko, M.A. (2002). Bone patterning is altered in the regenerating zebrafish caudal fin after ectopic expression of sonic hedgehog and *bmp2b* or exposure to cyclopamine. *Proc Natl Acad Sci U S A* *99*, 8713-8718.
- Rogers, K.W., Lord, N.D., Gagnon, J.A., Pauli, A., Zimmerman, S., Aksel, D.C., Reyon, D., Tsai, S.Q., Joung, J.K., and Schier, A.F. (2017). Nodal patterning without Lefty inhibitory feedback is functional but fragile. *Elife* *6*.
- Rohner, N., Bercsenyi, M., Orban, L., Kolanczyk, M.E., Linke, D., Brand, M., Nusslein-Volhard, C., and Harris, M.P. (2009). Duplication of *fgfr1* permits Fgf signaling to serve as a target for selection during domestication. *Curr Biol* *19*, 1642-1647.
- Schulte-Merker, S., Lee, K.J., McMahon, A.P., and Hammerschmidt, M. (1997). The zebrafish organizer requires *chordino*. *Nature* *387*, 862-863.
- Shiota, M., Hikita, Y., Kawamoto, Y., Kusakabe, H., Tanaka, M., Izumi, Y., Nakao, T., Miura, K., Funae, Y., and Iwao, H. (2012). Pravastatin-induced proangiogenic effects depend upon extracellular FGF-2. *J Cell Mol Med* *16*, 2001-2009.
- Steinbicker, A.U., Sachidanandan, C., Vonner, A.J., Yusuf, R.Z., Deng, D.Y., Lai, C.S., Rauwerdink, K.M., Winn, J.C., Saez, B., Cook, C.M., *et al.* (2011). Inhibition of bone morphogenetic protein signaling attenuates anemia associated with inflammation. *Blood* *117*, 4915-4923.
- Sun, L., Tran, N., Liang, C., Tang, F., Rice, A., Schreck, R., Waltz, K., Shawver, L.K., McMahon, G., and Tang, C. (1999). Design, synthesis, and evaluations of substituted 3-[(3-

4-carboxyethylpyrrol-2-yl)methylidene]indolin-2-ones as inhibitors of VEGF, FGF, and PDGF receptor tyrosine kinases. *J Med Chem* *42*, 5120-5130.

Takayama, K., Muto, A., and Kikuchi, Y. (2018). Leucine/glutamine and v-ATPase/lysosomal acidification via mTORC1 activation are required for position-dependent regeneration. *Sci Rep* *8*, 8278.

Terashima, A.V., Mudumana, S.P., and Drummond, I.A. (2014). Odd skipped related 1 is a negative feedback regulator of nodal-induced endoderm development. *Dev Dyn* *243*, 1571-1580.

Tuazon, F.B., Wang, X., Andrade, J.L., Umulis, D., and Mullins, M.C. (2020). Proteolytic Restriction of Chordin Range Underlies BMP Gradient Formation. *Cell Rep* *32*, 108039.

van Boxtel, A.L., Economou, A.D., Heliot, C., and Hill, C.S. (2018). Long-Range Signaling Activation and Local Inhibition Separate the Mesoderm and Endoderm Lineages. *Dev Cell* *44*, 179-191 e175.

Wang, X., Moon, J., Dodge, M.E., Pan, X., Zhang, L., Hanson, J.M., Tuladhar, R., Ma, Z., Shi, H., Williams, N.S., *et al.* (2013). The development of highly potent inhibitors for porcupine. *J Med Chem* *56*, 2700-2704.

Williams, B.B., Cantrell, V.A., Mundell, N.A., Bennett, A.C., Quick, R.E., and Jessen, J.R. (2012). VANGL2 regulates membrane trafficking of MMP14 to control cell polarity and migration. *J Cell Sci* *125*, 2141-2147.

Williams, M.L., and Solnica-Krezel, L. (2020). Nodal and planar cell polarity signaling cooperate to regulate zebrafish convergence and extension gastrulation movements. *Elife* *9*.

Yamashita, A., Morioka, M., Kishi, H., Kimura, T., Yahara, Y., Okada, M., Fujita, K., Sawai, H., Ikegawa, S., and Tsumaki, N. (2014). Statin treatment rescues FGFR3 skeletal dysplasia phenotypes. *Nature* *513*, 507-511.

Zinck, N.W., Jeradi, S., and Franz-Odenaal, T.A. (2021). Elucidating the early signaling cues involved in zebrafish chondrogenesis and cartilage morphology. *J Exp Zool B Mol Dev Evol* *336*, 18-31.

Decision Letter, first revision:

Dear Patrick,

Thank you for submitting your revised manuscript "EmbryoNet: Using deep learning to link embryonic phenotypes to signaling pathways" (NMETH-RS50482A). It has now been seen by the original referees and their comments are below. The reviewers find that the paper has improved in revision, and therefore we'll be happy in principle to publish it in Nature Methods, pending minor revisions to comply with our editorial and formatting guidelines.

TRANSPARENT PEER REVIEW

Nature Methods offers a transparent peer review option for new original research manuscripts submitted from 17th February 2021. We encourage increased transparency in peer review by publishing the reviewer comments, author rebuttal letters and editorial decision letters if the authors agree. Such peer review material is made available as a supplementary peer review file. Please state in the cover letter 'I wish to participate in transparent peer review' if you want to opt in, or 'I do not wish to participate in transparent peer review' if you don't. Failure to state your preference will result in delays in accepting your manuscript for publication.

ORCID

Sincerely,
Rita

Rita Strack, Ph.D.
Senior Editor
Nature Methods

Reviewer #1 (Remarks to the Author):

This revised version did appropriately respond to all my comments.

Marc Muller

Reviewer #2 (Remarks to the Author):

I find that the responses to my queries are comprehensive and make the paper suitable for publication in Nature Methods. I look forward to seeing it in print.

Final Decision Letter:

Dear Patrick,

I am pleased to inform you that your Resource, "EmbryoNet: Using deep learning to link embryonic phenotypes to signaling pathways", has now been accepted for publication in Nature Methods. Your paper is tentatively scheduled for publication in our June print issue, and will be published online prior to that. The received and accepted dates will be September 26, 2022 and April 5, 2023. This note is intended to let you know what to expect from us over the next month or so, and to let you know where to address any further questions.

Acceptance is conditional on the data in the manuscript not being published elsewhere, or announced in the print or electronic media, until the embargo/publication date. These restrictions are not intended to

deter you from presenting your data at academic meetings and conferences, but any enquiries from the media about papers not yet scheduled for publication should be referred to us.

Once your paper is typeset, you will receive an email with a link to choose the appropriate publishing options for your paper and our Author Services team will be in touch regarding any additional information that may be required.

Please note that *Nature Methods* is a Transformative Journal (TJ). Authors may publish their research with us through the traditional subscription access route or make their paper immediately open access through payment of an article-processing charge (APC). Authors will not be required to make a final decision about access to their article until it has been accepted. [Find out more about Transformative Journals](https://www.springernature.com/gp/open-research/transformative-journals)

Your paper will now be copyedited to ensure that it conforms to Nature Methods style. Once proofs are generated, they will be sent to you electronically and you will be asked to send a corrected version within 24 hours. It is extremely important that you let us know now whether you will be difficult to contact over the next month. If this is the case, we ask that you send us the contact information (email, phone and fax) of someone who will be able to check the proofs and deal with any last-minute problems.

If, when you receive your proof, you cannot meet the deadline, please inform us at rjsproduction@springernature.com immediately.

Once your manuscript is typeset and you have completed the appropriate grant of rights, you will receive a link to your electronic proof via email with a request to make any corrections within 48 hours. If, when you receive your proof, you cannot meet this deadline, please inform us at rjsproduction@springernature.com immediately.

Once your paper has been scheduled for online publication, the Nature press office will be in touch to confirm the details.

Once your paper has been scheduled for online publication, the Nature press office will be in touch to confirm the details.

Content is published online weekly on Mondays and Thursdays, and the embargo is set at 16:00 London time (GMT)/11:00 am US Eastern time (EST) on the day of publication. If you need to know the exact publication date or when the news embargo will be lifted, please contact our press office after you have submitted your proof corrections. Now is the time to inform your Public Relations or Press Office about your paper, as they might be interested in promoting its publication. This will allow them time to prepare an accurate and satisfactory press release. Include your manuscript tracking number NMETH-RS50482B and the name of the journal, which they will need when they contact our office.

About one week before your paper is published online, we shall be distributing a press release to news organizations worldwide, which may include details of your work. We are happy for your institution or funding agency to prepare its own press release, but it must mention the embargo date and Nature Methods. Our Press Office will contact you closer to the time of publication, but if you or your Press Office have any inquiries in the meantime, please contact press@nature.com.

Nature Portfolio journals [encourage authors to share their step-by-step experimental protocols](https://www.nature.com/nature-research/editorial-policies/reporting-standards#protocols) on a protocol sharing platform of their choice. Nature Portfolio 's Protocol Exchange is a free-to-use and open resource for protocols; protocols deposited in Protocol Exchange are citable and can be linked from the published article. More details can found at www.nature.com/protocolexchange/about.

Please note that you and any of your coauthors will be able to order reprints and single copies of the issue containing your article through Nature Portfolio 's reprint website, which is located at <http://www.nature.com/reprints/author-reprints.html>. If there are any questions about reprints please send an email to author-reprints@nature.com and someone will assist you.

Best regards,
Rita

Rita Strack, Ph.D.
Senior Editor
Nature Methods